# INFLUENCE-GUIDED DIFFUSION FOR DATASET DISTILLATION

**Mingyang Chen[1,2], Jiawei Du[3], Bo Huang[1,2], Yi Wang[4], Xiaobo Zhang[5], Wei Wang[1,2]**

[1] The Hong Kong University of Science and Technology (Guangzhou)
[2] The Hong Kong University of Science and Technology
[3] CFAR, A*STAR, Singapore
[4] Dongguan University of Technology
[5] Southwest Jiaotong University
mchenbt@connect.ust.hk, weiwcs@ust.hk

## ABSTRACT

Dataset distillation aims to streamline the training process by creating a compact yet effective dataset for a much larger original dataset. However, existing methods often struggle with distilling large, high-resolution datasets due to prohibitive resource costs and limited performance, primarily stemming from sample-wise optimizations in the pixel space. Motivated by the remarkable capabilities of diffusion generative models in learning target dataset distributions and controllably sampling high-quality data tailored to user needs, we propose framing dataset distillation as a controlled diffusion generation task aimed at generating data specifically tailored for effective training purposes. By establishing a correlation between the overarching objective of dataset distillation and the trajectory influence function, we introduce the Influence-Guided Diffusion (IGD) sampling framework to generate training-effective data without the need to retrain diffusion models. An efficient guided function is designed by leveraging the trajectory influence function as an indicator to steer diffusions to produce data with influence promotion and diversity enhancement. Extensive experiments show that the training performance of distilled datasets generated by diffusions can be significantly improved by integrating with our IGD method and achieving state-of-the-art performance in distilling ImageNet datasets. Particularly, an exceptional result is achieved on the ImageNet-1K, reaching 60.3% at IPC=50. Our code is available at `https://github.com/mchen725/DD_IGD`.

## 1 INTRODUCTION

Dataset distillation has gained significant attention due to its ability to balance the conflict demands of maintaining training effectiveness while overwhelming resource overhead. This method involves crafting a compact yet effective surrogate dataset for a large-scale original dataset. The surrogate is optimized to retain essential information from the cumbersome original, enabling models trained on it to achieve performance comparable to those trained on the complete one.

Early dataset distillation methods have made significant strides in distillation efficacy through various insightful paradigms (Zhao et al., 2021; Kim et al., 2022; Nguyen et al., 2021; Cazenavette et al., 2022; Du et al., 2023; Cui et al., 2023). However, their success is mainly limited to distilling small datasets like CIFAR (Krizhevsky & Hinton, 2009) or downscaled ImageNet (Russakovsky et al., 2015) with low resolution. Extending these methods to higher-resolution datasets (e.g., $\geq 128 \times 128$) is hindered by treating data as a entity and refining it at the pixel level. This escalates time and computational costs with data dimensionality and preset compression ratios, typically indicated by Images Per Class (IPC). Moreover, prioritizing pixel-level optimization overlooks distributional shifts from the original dataset. Yet, at higher resolutions, synthetic data retains ineffective high-frequency patterns, leading to performance degradation (Cazenavette et al., 2023).

Recognizing the robust capability to capture intricate data distributions, a recent approach (Gu et al., 2024) integrates diffusion models to tackle the high-resolution challenges faced by previous

pixel-oriented methods, achieving cutting-edge performance. This technique entails fine-tuning a latent diffusion model through a minimax criterion, yielding distilled datasets that harmonize representativeness and diversity for better alignment with the authentic data distribution. However, research on core-set selection techniques (Killamsetty et al., 2021a;b; Iyer et al., 2021) indicates that even data sampled directly from the authentic distribution can contribute unevenly to model training. Concerns remain about the effectiveness of the proposed objective in generating distilled datasets that are optimally tailored for highly effective training.

In this work, we introduce a new paradigm of using diffusion models in the task of dataset distillation, termed the **I**nfluence-**G**uided **D**iffusion (**IGD**) sampling method. This method is conceptually tailored to directly guide diffusion models in generating data under a generalized training-effective condition, without requiring to retrain diffusion models. We highlight the challenges inherent in achieving this target, particularly due to the abstract nature of the tailored condition, in contrast to existing controlled diffusion generation tasks that involve explicit content specifications (Rombach et al., 2022; Ho & Salimans, 2022). To address this challenge, we first establish a correlation between the overarching objective of dataset distillation and the trajectory influence function (Pruthi et al., 2020), which approximates the impact of training on given data in terms of test loss

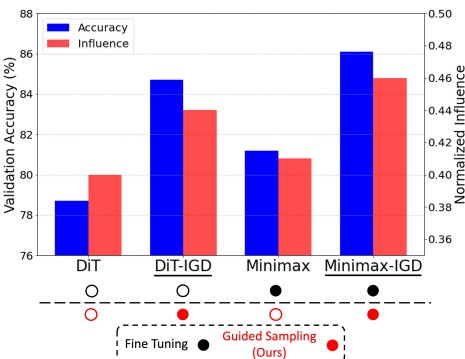

Figure 1: Enhanced cross-architecture performance with average influence by integrating IGD in distilling ImageNette with IPC=100.

variation. Building on this connection, we develop diversity-constraint and influence-based guidance as indicators to steer the diffusion models towards generating data with influence promotion and diversity enhancement, as illustrated in Figure 5. As evidenced by Figure 1, integrating IGD significantly enhances the performance of the vanilla Diffusion Transformer (DiT), outperforming results obtained through the fine-tuning method Minimax. Moreover, IGD complements Minimax to achieve even better results, with a simultaneous increase in influence.

In summary, our contributions are as follows:

- We propose a new scheme for dataset distillation by framing the task as a guided-diffusion generation problem.
- We establish a novel diffusion sampling framework that pioneers the integration of the influence function as a guidance for the controlled diffusion generation, with the aim of achieving generalized training-enhancing objectives.
- Experimental results illustrate that our method significantly improves the performance of diffusion models across different architectures on two ImageNet subsets. Furthermore, a state-of-the-art result is achieved on the ImageNet-1K, reaching 60.3% at IPC=50.

## 2 PRELIMINARIES

### 2.1 BACKGROUND ON DATASET DISTILLATION

We refer to the target dataset as $\mathcal{T} = \{(\boldsymbol{x}_i, \boldsymbol{y}_i)\}_{i=1}^{|\mathcal{T}|}$. Each sample $\boldsymbol{x}_i$ is drawn i.i.d. from a natural distribution $q(\boldsymbol{x})$, where $\boldsymbol{x}_i \in \mathbb{R}^d$ and $\boldsymbol{y}_i \in \mathcal{Y} = \{1, 2, \ldots, C\}$ refers to the ground-truth label. Dataset Distillation (DD) aims to condense this large labelled dataset $\mathcal{T}$ into a smaller synthetic dataset $\mathcal{S} = \{(\boldsymbol{u}_i, \boldsymbol{y}_i)\}_{i=1}^{|\mathcal{S}|}$, with $\boldsymbol{u}_i \in \mathbb{R}^d$ and $\boldsymbol{y}_i \in \mathcal{Y}$, such that $|\mathcal{S}| \ll |\mathcal{T}|$. The reduced dataset $\mathcal{S}$ is optimized to retain essential information from $\mathcal{T}$ to ensure that any model initialized with parameters $\boldsymbol{\theta}_0$ can be optimized to minimize the validation loss on the target dataset $\mathcal{T}$:

$$\min_{\mathcal{S}} \frac{1}{|\mathcal{T}|} \sum_{i=1}^{|\mathcal{T}|} \left[ \ell\left(\boldsymbol{x}_i, \boldsymbol{y}_i; \boldsymbol{\theta}_*^{\mathcal{S}}\right) - \ell\left(\boldsymbol{x}_i, \boldsymbol{y}_i; \boldsymbol{\theta}_0\right) \right] \quad s.t. \; \boldsymbol{\theta}_*^{\mathcal{S}} = Alg(\mathcal{S}, \boldsymbol{\theta}_0). \quad (1)$$

Here, $Alg(\mathcal{S}, \boldsymbol{\theta}_0) = \arg\min_{\boldsymbol{\theta}} \mathbb{E}_{(\boldsymbol{u}_i, \boldsymbol{y}_i) \in \mathcal{S}} \left[\ell(\boldsymbol{u}_i, \boldsymbol{y}_i; \boldsymbol{\theta})\right]$ represents the training algorithm that optimizes the initialized parameters $\boldsymbol{\theta}_0$ over the synthetic data $\mathcal{S}$, and $\ell(\boldsymbol{x}, \boldsymbol{y}; \boldsymbol{\theta})$ denotes the prediction

loss of a model with parameters $\boldsymbol{\theta}$ on a data pair $(\boldsymbol{x}, \boldsymbol{y})$. To prevent unexpected distributional shift, we propose to frame DD as *learning a conditional distribution of the authentic distribution*, e.g., $p(\boldsymbol{x}|Condition)$, to sample near-real data under the generalized training-effective conditions.

## 2.2 GUIDED DIFFUSION GENERATION

Given samples from the data distribution $q(\boldsymbol{x})$, diffusion models are capable of learning a parameterized distribution $p_\phi(\boldsymbol{x})$ that approximates $q(\boldsymbol{x})$ and is easy to sample from it (Song et al., 2020b). On a high level, this is implemented through a forward noising process and a reverse denoising process. Concretely, the forward process gradually adds Gaussian noise $\boldsymbol{\epsilon} \sim \mathcal{N}(0, I)$ of different magnitudes to clean data point $\boldsymbol{z}_0$: $\boldsymbol{z}_t = \sqrt{\alpha_t}\boldsymbol{z}_0 + \sqrt{1-\alpha_t}\boldsymbol{\epsilon}$, where $\alpha_t$ controls the noise scale at step $t$. A diffusion model is a denoising function that learns by minimizing the dissimilarity, e.g., mean squared error, between the predicted noise $\boldsymbol{\epsilon}_\phi(\boldsymbol{z}_t, t, c)$ and $\boldsymbol{\epsilon}$, where $c$ is a conditional input such as labels. The reverse process generates denoised samples by sampling from $p_\phi(\boldsymbol{z}_{t-1}|\boldsymbol{z}_t, \boldsymbol{z}_0)$, which is generally parameterized as a Gaussian distribution and varies across studies in its approximation (Ho et al., 2020). For instance, Denoising Diffusion Implicit Model (DDIM) (Song et al., 2020a) first predicts the clean data point $\hat{\boldsymbol{z}}_{0|t}$ based on $\boldsymbol{z}_t$ as:

$$\hat{\boldsymbol{z}}_{0|t} = \frac{1}{\sqrt{\alpha_t}}(\boldsymbol{z}_t - \sqrt{1-\alpha_t}\boldsymbol{\epsilon}_\phi(\boldsymbol{z}_t, t, c)). \tag{2}$$

$\boldsymbol{z}_{t-1}$ is then sampled from $\mathcal{N}\left(\sqrt{\alpha_{t-1}}\hat{\boldsymbol{z}}_{0|t} + \sqrt{1-\alpha_{t-1}-\sigma_t^2}\boldsymbol{\epsilon}_\phi(\boldsymbol{z}_t, t, c), \sigma_t^2 I\right)$, where $\sigma_t$ is the predefined noise factor. For notation simplicity, we abstract this process as: $\boldsymbol{z}_{t-1} = s(\boldsymbol{z}_t, t, \boldsymbol{\epsilon}_\phi)$. In this work, we adopt the widely utilized *latent diffusion* (Peebles & Xie, 2023) as the backbone. Here, an encoder is employed to transform images to latent codes $\boldsymbol{z} = E(\boldsymbol{x})$ and a decoder $D(\cdot)$ reconstructs latent codes back to the image space to obtain the distilled dataset $\mathcal{S} = \{(D(\boldsymbol{z}_i), \boldsymbol{y}_i)\}_{i=1}^{|\mathcal{S}|}$.

Diffusion models typically employ conditioning to tailor outputs to specific user inputs, such as labels or text prompts. However, our purpose diverges from explicit content specifications, focusing instead on more abstract requirements. We aim to guide the diffusion model to identify conditional distributions within the learned distribution and selectively sample data to optimize training effectiveness. To this end, we employ a more adaptable method of controlling model outputs through *guided-diffusion generation* (Bansal et al., 2023; Yu et al., 2023; Gopalakrishnan Nair et al., 2023). These methods are largely inspired by the energy-based model (EBM) used for formulating score-based diffusions (Song et al., 2020b; 2021). Intuitively, any metric function $f_C(.)$ that subtly measures the compatibility of the noisy sample $\boldsymbol{z}_t$ to the condition $C$ is valid for providing steering guidance. By this means, the sampling step can generally be implemented as:

$$\boldsymbol{z}_{t-1} = s(\boldsymbol{z}_t, t, \boldsymbol{\epsilon}_\phi) - \rho_t \nabla_{\boldsymbol{z}_t} f_C(\boldsymbol{z}_t), \tag{3}$$

where $\rho_t$ is defined to align with the denoising scale of the current $\boldsymbol{\epsilon}_\phi(\boldsymbol{z}_t, t)$. By introducing a meticulously designed guided function that effectively measures the impact of data on training efficacy (e.g., as depicted by validation loss), this implementation seamlessly aligns with our objective of framing dataset distillation as sampling data from a desirable conditional distribution.

## 3 METHOD

### 3.1 ESTIMATING DATA INFLUENCE AS DIFFUSION CONDITIONAL GUIDANCE

We identify influence function (Koh & Liang, 2017) as insightful parallel research that can quantify the impact of specific training data on model validation loss. This is highly relevant to the design of metric functions used for steering guidance in diffusion models under our training-effective condition. Leveraging the Fundamental Theorem of Calculus, Pruthi et al. (2020) introduced trajectory influence to estimate the cumulative influence of a training data pair $(\boldsymbol{x}, \boldsymbol{y})$ on validation data pair $(\boldsymbol{x}', \boldsymbol{y}')$. This method integrates the stepwise changes in the loss of the validation data throughout the training process. In our case, employing Stochastic Gradient Descent (SGD) as the training algorithm $Alg$, the model update can be expressed as $\boldsymbol{\theta}_{t+1} - \boldsymbol{\theta}_t = -\eta_t \nabla_{\boldsymbol{\theta}} \ell(\boldsymbol{x}, \boldsymbol{y}; \boldsymbol{\theta}_t)$, where $\eta_t$ represents the learning rate at timestep $t$. Utilizing the first-order Taylor expansion, the loss change of $(\boldsymbol{x}', \boldsymbol{y}')$ at

each timestep can be approximated by:

$$\ell(\boldsymbol{x}', \boldsymbol{y}'; \boldsymbol{\theta}_{t+1}) - \ell(\boldsymbol{x}', \boldsymbol{y}'; \boldsymbol{\theta}_t) \approx \nabla_{\boldsymbol{\theta}}\ell(\boldsymbol{x}', \boldsymbol{y}'; \boldsymbol{\theta}_t) \cdot (\boldsymbol{\theta}_{t+1} - \boldsymbol{\theta}_t)$$
$$= -\eta_t \nabla_{\boldsymbol{\theta}}\ell(\boldsymbol{x}', \boldsymbol{y}'; \boldsymbol{\theta}_t) \cdot \nabla_{\boldsymbol{\theta}}\ell(\boldsymbol{x}, \boldsymbol{y}; \boldsymbol{\theta}_t). \tag{4}$$

The overall influence of $(\boldsymbol{x}, \boldsymbol{y})$ on $(\boldsymbol{x}', \boldsymbol{y}')$ throughout the training trajectory is quantified by aggregating these stepwise changes across epochs:

$$\mathcal{I}(\boldsymbol{x}, \boldsymbol{x}') = \sum_{e=0}^{E} \bar{\eta}_e \nabla_{\boldsymbol{\theta}}\ell\left(\boldsymbol{x}', \boldsymbol{y}'; \boldsymbol{\theta}_e\right) \cdot \nabla_{\boldsymbol{\theta}}\ell\left(\boldsymbol{x}, \boldsymbol{y}; \boldsymbol{\theta}_e\right) \propto \ell(\boldsymbol{x}', \boldsymbol{y}'; \boldsymbol{\theta}_0) - \ell(\boldsymbol{x}', \boldsymbol{y}'; \boldsymbol{\theta}_E), \tag{5}$$

where $\bar{\eta}_e$ denotes the learning rate of the $e$-th epoch, for a total of $E$ epochs. By substituting the validation data $(\boldsymbol{x}', \boldsymbol{y}')$ as the real data in the original dataset, this formulation is an effective approximation to the general objective of dataset distillation defined in Equation (1). Based on this insight, we define the objective of the guidance for a latent code $z$ given a certain class $c$ as:

$$\max_{\boldsymbol{z}} \frac{1}{|\mathcal{T}_c|} \sum_{i=1}^{|\mathcal{T}_c|} \mathcal{I}(D(\boldsymbol{z}), \boldsymbol{x}_i) = \max_{\boldsymbol{z}} \sum_{e=0}^{E} \bar{\eta}_e \bar{\nabla}_{\boldsymbol{\theta}}\ell_c\left(\boldsymbol{\mathcal{X}}_c; \boldsymbol{\theta}_e^{\mathcal{S}}\right) \cdot \nabla_{\boldsymbol{\theta}}\ell_c\left(D(\boldsymbol{z}); \boldsymbol{\theta}_e^{\mathcal{S}}\right), \tag{6}$$

where $\bar{\nabla}_{\boldsymbol{\theta}}\ell_c(\boldsymbol{\mathcal{X}}_c; \boldsymbol{\theta}_e^{\mathcal{S}}) = \frac{1}{|\mathcal{T}_c|} \sum_{i=1}^{|\mathcal{T}_c|} \nabla_{\boldsymbol{\theta}}\ell\left(\boldsymbol{x}_i, c; \boldsymbol{\theta}_e^{\mathcal{S}}\right)$ based on the Fubini's Theorem and $\mathcal{T}_c$ is the subset of the given class $c$, $\boldsymbol{\theta}_e^{\mathcal{S}}$ represents a checkpoint obtained on the decoded data. Intuitively, this objective can be optimized if models trained on synthetic data obtain trajectories equivalent to those trained on $\mathcal{T}_c$, thereby maximizing the validation loss drop. This essentially shares a similar purpose with the Gradient-Matching (GM) scheme (Zhao et al., 2021; Zhao & Bilen, 2021a). However, we identify three primary issues with directly adapting this formulation as the metric function for guided diffusion in dataset distillation: (1) prohibitive cost: the necessity of model retraining at each diffusion sampling step is computationally burdensome; (2) accumulated error: akin to the limitations of the GM method, the gap between trajectories inevitably accumulates during training on synthetic data, leading to ineffective matching and consequently degraded performance (Cazenavette et al., 2022); (3) information redundancy: the relatively poor diversity of diffusion-generated data limits its effectiveness for dataset distillation (Du et al., 2023), and matching with the averaged real gradients, as shown in Equation (6), may further exacerbate this issue.

In the following section, we tackle these challenges by developing diversity-constrained guided functions and detailing our **I**nfluence-**G**uided **D**iffusion (**IGD**) sampling framework.

## 3.2 Efficient Influence-Guided Diffusion Sampling with Diversity Constraint

Denote $\boldsymbol{\theta}_e^{\mathcal{T}_c} = \boldsymbol{\theta}_{e-1}^{\mathcal{T}_c} - \bar{\eta}_{e-1}\bar{\nabla}_{\boldsymbol{\theta}}\ell_c(\boldsymbol{\mathcal{X}}_c; \boldsymbol{\theta}_{e-1}^{\mathcal{T}_c})$ as checkpoints trained on the real subset $\mathcal{T}_c$ with SGD and the same learning rate schedule as on the synthetic data. Replacing the checkpoints $\boldsymbol{\theta}_e^{\mathcal{S}}$ with $\boldsymbol{\theta}_e^{\mathcal{T}_c}$ in Equation (6) is an optimizably equivalent target. This equivalence holds because these two targets converge to the same optimal solution when $z$ can provide the same training dynamics as $\mathcal{T}_c$, i.e., $\bar{\nabla}_{\boldsymbol{\theta}}\ell_c(\boldsymbol{\mathcal{X}}_c; \boldsymbol{\theta}_e^{\mathcal{T}_c}) = \nabla_{\boldsymbol{\theta}}\ell_c(D(\boldsymbol{z}); \boldsymbol{\theta}_e^{\mathcal{T}_c})\ \forall e \in [0, E]$. Building on this insight, in practical implementation, we extend this usage to the checkpoints $\boldsymbol{\theta}_e^{\mathcal{T}}$ obtained through standard mini-batch updates over the entire dataset $\mathcal{T}$. This adjustment mitigates the mismatch caused by the discrepancy between synthetic and real trajectories (Kim et al., 2022), while also eliminating the time cost associated with retraining models on $\mathcal{S}$ at each sampling step. Additionally, we use cosine similarity instead of the dot product to stabilize the magnitude of the guidance signal provided by the influence function. These modifications yield the influence guided loss function as:

$$\mathcal{G}_I(\boldsymbol{z}) = \frac{1}{|E|} \sum_{e=1}^{E} \bar{\eta}_e \left(1 - \frac{\bar{\nabla}_{\boldsymbol{\theta}}\ell_c\left(\boldsymbol{\mathcal{X}}_c; \boldsymbol{\theta}_e^{\mathcal{T}}\right) \cdot \nabla_{\boldsymbol{\theta}}\ell_c\left(D(z); \boldsymbol{\theta}_e^{\mathcal{T}}\right)}{\left\|\bar{\nabla}_{\boldsymbol{\theta}}\ell_c\left(\boldsymbol{\mathcal{X}}_c; \boldsymbol{\theta}_e^{\mathcal{T}}\right)\right\| \left\|\nabla_{\boldsymbol{\theta}}\ell_c\left(D(z); \boldsymbol{\theta}_e^{\mathcal{T}}\right)\right\|}\right). \tag{7}$$

Directly computing the influence over an intermediate noisy $\boldsymbol{z}_t$ is undesirable, as the checkpoints $\boldsymbol{\theta}_e^{\mathcal{T}}$ are trained on clean data and do not adapt to provide meaningful guidance as a metric function when the input is noisy (Ho & Salimans, 2022). To mitigate this issue, we utilize the *predicted* clean data $\hat{\boldsymbol{z}}_{0|t}$ of the current $\boldsymbol{z}_t$, based on Equation (2) as defined by DDIM, as an approximation of the real $\boldsymbol{z}_0$. Subsequently, we compute the influence guidance $\mathcal{G}_I(\hat{\boldsymbol{z}}_{0|t})$ on the predicted clean data and derive the guided gradient $\nabla_{\boldsymbol{z}_t}\mathcal{G}_I((\boldsymbol{z}_t - \sqrt{1 - \alpha_t}\boldsymbol{\epsilon}_\phi(\boldsymbol{z}_t, t))/\sqrt{\alpha_t})$ through backpropagation.

---

**Algorithm 1:** Influence-Guided Diffusion Sampling

---

1  **Parameters:** Class $c$, influence factor $\rho_t$, deviation factor $\gamma_t$, scales $\{\alpha_t\}_{t=1}^{T}$, guided range $A,B$

2  **Required:** Pre-trained diffusion model $\boldsymbol{\epsilon}_\phi$, list of retained checkpoints $\mathcal{R}$, list of averaged gradients $G_c$, generated data memory $\mathcal{M}_c$, decoder model $D$

3  **Initialize:** Sample initial random noise $\boldsymbol{z}_T \sim \mathcal{N}(0, I)$;

4  **for** $t = T$ **to** $1$ **do**

5     Obtain the denoised signal $\boldsymbol{\epsilon}_\phi(\boldsymbol{z}_t, t, c)$ from the diffusion model;

6     **if** $t$ *in* $[A, B]$ **then**

7         Calculate the influence metric $\mathcal{G}_I(\hat{\boldsymbol{z}}_{0|t})$ as Equation (7) with $\mathcal{R}$ and $G_c$;

8         Calculate the deviation metric $\mathcal{G}_D(\boldsymbol{z}_t)$ as Equation (8) with $\mathcal{M}_c$;

9         Implement guided sampling $\boldsymbol{z}_{t-1} = s(\boldsymbol{z}_t, t, \boldsymbol{\epsilon}_\phi) - \rho_t \nabla_{\boldsymbol{z}_t} \mathcal{G}_I(\hat{\boldsymbol{z}}_{0|t}) - \gamma_t \nabla_{\boldsymbol{z}_t} \mathcal{G}_D(\boldsymbol{z}_t)$;

10    **else**

11       Implement vanilla sampling $\boldsymbol{z}_{t-1} = s(\boldsymbol{z}_t, t, \boldsymbol{\epsilon}_\phi)$;

12  **return** Decoded synthetic image $D(\boldsymbol{z}_0)$;

---

To ensure diversity and avoid excessive redundancy in the surrogate dataset's training signals, we propose adding a constraint to the generation objective. This constraint ensures that the similarity between generated data within a certain class does not exceed a specified threshold: $\text{sim}(\boldsymbol{z}_i, \boldsymbol{z}_j) \leq \delta, \ \forall \boldsymbol{z}_i, \boldsymbol{z}_j \in \mathcal{Z}_c$, where $\boldsymbol{z}_i \neq \boldsymbol{z}_j$. In practice, we incorporate this constraint using a Lagrangian multiplier and propose a deviation guidance function to optimize it in each guided sampling step:

$$\mathcal{G}_D(\boldsymbol{z}) = \frac{\boldsymbol{z} \cdot \tilde{\boldsymbol{z}}^*}{\|\boldsymbol{z}\|\|\tilde{\boldsymbol{z}}^*\|} \quad \text{subject to} \quad \tilde{\boldsymbol{z}}^* = \arg\max_{\tilde{\boldsymbol{z}} \in \mathcal{M}^c} \frac{\boldsymbol{z} \cdot \tilde{\boldsymbol{z}}}{\|\boldsymbol{z}\|\|\tilde{\boldsymbol{z}}\|}, \tag{8}$$

where $\mathcal{M}^c$ represents the set of all previously generated data for a certain class $c$.

Ultimately, we utilize the influence guidance of $\mathcal{G}_I(\hat{\boldsymbol{z}}_{0|t})$ alongside the deviation guidance of $\mathcal{G}_D(\boldsymbol{z}_t)$, reformulating the guided sampling step as:

$$\boldsymbol{z}_{t-1} = s(\boldsymbol{z}_t, t, \boldsymbol{\epsilon}_\phi) - \rho_t \nabla_{\boldsymbol{z}_t} \mathcal{G}_I(\hat{\boldsymbol{z}}_{0|t}) - \gamma_t \nabla_{\boldsymbol{z}_t} \mathcal{G}_D(\boldsymbol{z}_t), \text{ where } \rho_t = k \cdot \sqrt{1 - \alpha_t} \frac{\|\boldsymbol{\epsilon}_\phi(\boldsymbol{z}_t, t, c)\|}{\|\nabla_{\boldsymbol{z}_t} \mathcal{G}_I(\hat{\boldsymbol{z}}_{0|t})\|} \tag{9}$$

is the scale factor designed to adaptively adjust the magnitude of the influence guidance alongside the dynamics of the denoised signal $\boldsymbol{\epsilon}_\phi$, and $\gamma_t$ is empirically preset for the deviation guidance. Furthermore, we introduce two practical techniques that are essential for enhancing both the efficiency and efficacy of the proposed IGD framework.

**Choosing representative checkpoints via gradient similarity.** For efficiency, trajectory influence initially suggests saving checkpoints at regular intervals to compute step-wise influence. However, given the non-linear nature of training dynamics, evenly spaced checkpoints may scatter attention to critical stages. To efficiently calculate the influence guidance, we propose a simple yet effective filtering algorithm. We store $\boldsymbol{\theta}_0^{\mathcal{T}}$ as the first checkpoint in a list $\mathcal{R}$ and compute its averaged gradient $\mathbb{E}_c[\nabla_{\boldsymbol{\theta}} \ell_c(\boldsymbol{\mathcal{X}}_c; \boldsymbol{\theta}_0^{\mathcal{T}})]$ as the initial reference. For each subsequent checkpoint, we compute the averaged gradient and calculate its cosine similarity with the reference. If the similarity is below a given threshold, we store the current checkpoint and update its averaged gradient as the new reference. This process traverses all epochs, and only the retained checkpoints in $\mathcal{R}$ are used by influence guidance.

**Mitigating overfitting and reducing runtime by early-stage Guidance.** Guided diffusion tasks face a trade-off between generation quality and the impact of guidance (Lugmayr et al., 2022; Bansal et al., 2023). In our problem, we observe that samples generated with a large preset $k$ in $\rho_t$ achieve significant influence loss reduction but also exhibit noticeable abnormalities and degraded performance. Detailed evaluations are provided in Section 4.4. Empirical observations in diffusion generation demonstrate that most semantic content is generated during the early-to-mid stages of sampling (Yu et al., 2023). We adopt guided sampling only in these partial steps, allowing vanilla sampling to refine details in the remaining steps. For example, in DDIM with 50 sampling steps, guided sampling is applied only when $t$ is in $[30, 45]$. This approach allows data generated with strong guidance to maintain comparable influence without noticeable abnormalities or performance degradation. Consequently, this also reduces the runtime associated with guidance calculation.

Table 1: **ImageNette & ImageWoof:** Performance comparison with state-of-the-art pixel-level distillation methods, pretrained DiT and Minimax-tuned DiT models with vanilla generation. DiT-IGD and Minimax-IGD denote utilizing our proposed IGD sampling framework for generation.

| Dataset | Model | IPC | Random | DM | IDC-1 | DiT | DiT-IGD | Minimax | Minimax-IGD | Full |
|---------|-------|-----|--------|-----|-------|-----|---------|---------|-------------|------|
| Nette | ConvNet-6 | 10 | 46.0±0.5 | 49.8±1.1 | 48.2±1.2 | 56.2±1.3 | **61.9±1.9** | 58.2±0.9 | 58.8±1.0 | 94.3±0.5 |
| | | 50 | 71.8±1.2 | 70.3±0.8 | 72.4±0.7 | 74.1±0.6 | 80.9±0.9 | 76.9±0.9 | **82.3±0.8** | |
| | | 100 | 79.9±0.8 | 78.5±0.8 | 80.6±1.1 | 78.2±0.3 | 84.5±0.7 | 81.1±0.3 | **86.3±0.8** | |
| | ResNetAP-10 | 10 | 54.2±1.2 | 60.2±0.7 | 60.4±0.6 | 62.8±0.8 | **66.5±1.1** | 63.2±1.0 | 63.5±1.1 | 94.6±0.5 |
| | | 50 | 77.3±1.0 | 76.7±1.0 | 77.4±0.7 | 76.9±0.5 | 81.0±1.2 | 78.2±0.7 | **82.3±1.1** | |
| | | 100 | 81.1±0.6 | 80.9±0.7 | 81.5±1.2 | 80.1±1.1 | 85.2±0.5 | 81.3±0.9 | **86.1±0.9** | |
| | ResNet-18 | 10 | 55.8±1.0 | 60.9±0.7 | 61.0±0.8 | 62.5±0.9 | **67.7±0.3** | 64.9±0.6 | 66.2±1.2 | 95.3±0.6 |
| | | 50 | 75.8±1.1 | 75.0±1.0 | 77.8±0.7 | 75.2±0.9 | 81.0±0.7 | 78.1±0.6 | **82.0±0.3** | |
| | | 100 | 82.0±0.4 | 81.5±0.4 | 81.7±0.8 | 77.8±0.6 | 84.4±0.8 | 81.3±0.7 | **86.0±0.6** | |
| Woof | ConvNet-6 | 10 | 25.2±1.1 | 27.6±1.2 | 34.1±0.8 | 32.3±0.8 | 35.0±0.8 | 33.5±1.4 | **36.2±1.6** | 85.9±0.4 |
| | | 50 | 41.9±1.4 | 43.8±1.1 | 42.6±0.9 | 48.5±1.3 | 54.2±0.7 | 50.7±1.8 | **55.7±0.8** | |
| | | 100 | 52.3±1.5 | 50.1±0.9 | 51.0±1.1 | 54.2±1.5 | 61.1±1.0 | 57.1±1.9 | **63.0±1.8** | |
| | ResNetAP-10 | 10 | 31.6±0.8 | 29.8±1.0 | 38.5±0.7 | 39.0±0.9 | 41.0±0.8 | 39.6±1.2 | **43.3±0.3** | 87.2±0.6 |
| | | 50 | 50.1±1.6 | 47.8±1.2 | 49.3±0.9 | 55.8±1.1 | 62.7±1.2 | 59.8±0.8 | **65.0±0.8** | |
| | | 100 | 59.2±0.9 | 59.8±1.3 | 56.4±0.5 | 62.5±0.9 | 69.7±0.9 | 66.8±1.2 | **71.5±0.8** | |
| | ResNet-18 | 10 | 30.9±1.3 | 30.2±0.6 | 36.7±0.8 | 40.6±0.6 | 44.8±0.8 | 42.2±1.2 | **47.2±1.6** | 89.0±0.6 |
| | | 50 | 54.0±0.8 | 53.9±0.7 | 54.5±1.0 | 57.4±0.7 | 62.0±1.1 | 60.5±0.5 | **65.4±1.8** | |
| | | 100 | 63.6±0.5 | 64.9±0.7 | 57.7±0.8 | 62.3±0.5 | 70.6±1.8 | 67.4±0.7 | **72.1±0.9** | |

Algorithm 1 outlines the detailed process of our influence-guided diffusion sampling framework for generating each synthetic image. Before constructing the surrogate dataset, we first obtain checkpoints $\{\boldsymbol{\theta}_e^{\mathcal{T}}\}_{e=1}^{E}$ trained on $\mathcal{T}$ and apply the proposed filtering algorithm to retain representative checkpoints in the list $\mathcal{R}$. Before initiating generation for a specific class $c$, we calculate the averaged gradient $\bar{\nabla}_{\boldsymbol{\theta}}\ell_c(\boldsymbol{\mathcal{X}}_c; \boldsymbol{\theta}_e^{\mathcal{T}})$ across each retained checkpoint and store them in a list $G_c$. Subsequently, we execute the algorithm, storing the generated images in memory $\mathcal{M}_c$ until the desired number of images reaches the preset target IPC (images per class).

## 4 EXPERIMENTS

### 4.1 EXPERIMENTAL SETUP

**Datasets.** As our primary interest lies in large-scale, high-resolution distillation tasks, we assess the performance of our method on the complete ImageNet-1K dataset (224×224) (Russakovsky et al., 2015). To provide comparable evaluations across varying task difficulties, we conduct comprehensive experiments on two representative subsets, ImageNette and ImageWoof (Howard, 2019). ImageNette, consisting of 10 classes with less similarity and therefore easier to distinguish between, contrasts with ImageWoof, a challenging subset containing 10 classes of dog breeds.

**Baselines and evaluation metric.** We compare our method with several state-of-the-art dataset distillation methods including DM (Zhao & Bilen, 2021b), IDC-1 (Kim et al., 2022), SRe²L (Yin et al., 2024), G-VBSM (Shao et al., 2023), and RDED (Sun et al., 2024). Additionally, we regard pretrained DiT (Peebles & Xie, 2023) as a notable baseline because it achieves performance comparable to state-of-the-art methods even without tailored optimizations for dataset distillation. Furthermore, we include Minimax (Gu et al., 2024), a recent work refined DiT specifically for dataset distillation through a fine-tuning scheme, as a perpendicular baseline. Test architectures include ConvNet-6, ResNet-10 (He et al., 2016) with Average Pooling, ResNet-18, ResNet-101, MoblieNet-V2 (Sandler et al., 2018), EfficientNet-B0 (Tan, 2019) and Swin Transformer (Liu et al., 2021). The top-1 test accuracies of models trained on distilled datasets with different IPC (Image Per Class) are reported.

**Implementation detail.** For a fair comparison, we follow the official implementation of Minimax, utilizing a latent DiT model from Pytorch's official repository and an open-source VAE model from Stable Diffusion. DDIM (Song et al., 2020a) with 50 denoised steps is used as the vanilla sampling method for generation. For each test dataset, we train a 6-layer ConvNet (ConvNet-6) for 50 epochs with the learning rate $1 \times 10^{-2}$ to collect the surrogate checkpoints used in Equation (7). The similarity threshold for choosing representative checkpoints is set as 0.7. The detailed setup of hyperparameters $k$ and $\gamma_t$ for each datasets is discussed in Appendix A.10. All the experimental results of our method can be obtained on a single RTX 4090 GPU.

Table 2: **ImageNet-1K:** Performance comparison over ResNet-18 with state-of-the-art dataset distillation methods, pretrained DiT and Minimax-tuned DiT models with vanilla DDIM generation.

| Dataset | IPC | SRe$^2$L | G-VBSM | RDED | DiT | DiT-IGD | Minimax | Minimax-IGD |
|---|---|---|---|---|---|---|---|---|
| ImageNet-1K | 10 | 21.3±0.6 | 31.4±0.5 | 42.0±0.1 | 39.6±0.4 | 45.5±0.5 | 44.3±0.5 | **46.2±0.6** |
| | 50 | 46.8±0.2 | 51.8±0.4 | 56.5±0.1 | 52.9±0.6 | 59.8±0.3 | 58.6±0.3 | **60.3±0.4** |

Table 3: **ImageNet-1K:** Cross-architecture generalization performance comparison.

| | ResNet101 | | MobileNet-V2 | | EfficientNet-B0 | | Swin Transformer | |
|---|---|---|---|---|---|---|---|---|
| | IPC10 | IPC50 | IPC10 | IPC50 | IPC10 | IPC50 | IPC10 | IPC50 |
| RDED | 48.3±1.0 | 61.2±0.4 | **40.4±0.1** | 53.3±0.2 | 31.0±0.1 | 58.5±0.4 | 42.3±0.6 | 53.2±0.8 |
| DiT-IGD | 52.6±1.2 | 66.2±0.2 | 39.2±0.2 | 57.8±0.2 | 47.7±0.1 | 62.0±0.1 | 44.1±0.6 | **58.6±0.5** |
| Minimax-IGD | **53.4±0.9** | **66.8±0.2** | 39.7±0.4 | **58.5±0.3** | **48.5±0.1** | **62.7±0.2** | **44.8±0.8** | 58.2±0.5 |

## 4.2 COMPARISON WITH STATE-OF-THE-ART METHODS

**Evaluation on Woof & Nette.** As a training-free sampling framework, our IGD method can be incorporated into the pretrained DiT and Minimax-tuned DiT during the generation process. We designate these two methods as **DiT-IGD** and **Minimax-IGD**, respectively. As depicted in Table 1, our IGD-based methods demonstrate a significant improvement over the original backbone methods, and achieve state-of-the-art performance across both Woof and Nette datasets in all IPC settings. These enhancements are consistently observed across all evaluations conducted on the three tested architectures, highlighting a robust cross-architecture generalization ability. Particularly for IPC $\geq$ 50, DiT-IGD notably enhances the performance of DiT by 5.8% on Nette and by 6.6% on Woof, on average. Comparing with Minimax, Minimax-IGD averagely provides a 4.7% boost on Nette and a 5.1% boost on Woof. Moreover, we observed that DiT-IGD outperforms Minimax in most evaluations. Especially for the easier dataset Nette, despite the class distinctions facilitating knowledge condensation, Minimax only shows a marginal average improvement of 2.5% over DiT at IPC=100. In contrast, DiT-IGD achieves an average boost of 6.1%. Compared to diffusion-based methods, the pixel-level optimization methods DM and IDC-1 achieve moderate performance gains over random original images at IPC=10. However, as the IPC increases, the performance gain drastically diminishes or even becomes negative.

**Evaluation on ImageNet-1K.** Recent approaches proposed for efficiently distilling ImageNet-1K data rely on using well-trained models to provide synthetic images with soft labels to acquire richer information. Following the evaluation protocol of the RDED, we employ a ResNet-18 model, trained on the original dataset, to generate soft labels for synthetic images. The performances shown in Table 2 are evaluated over the same ResNet-18 architecture. The results demonstrate consistent improvements in integrating our IGD method over the DiT and Minimax methods. In particular, DiT-IGD demonstrates significant improvement to raw DiT, with enhancements of 5.9% at IPC=10 and 6.9% at IPC=50. This also positions our Minimax-IGD method at the forefront of this practical distillation task, surpassing the state-of-the-art image-based method RDED by 4.0%. In the **cross-architecture comparison** detailed in Table 3, synthetic datasets generated using our IGD methods generally outperform those created by RDED across four different unseen networks. Notably, our DiT-IGD and Minimax-IGD methods surpass RDED by an average margin of 4.6% and 5.0% at IPC=50, respectively. These remarkable performance improvements underline the promising potential of diffusion-based methods in the future of dataset distillation research.

## 4.3 CROSS-ARCHITECTURE ROBUSTNESS OF INFLUENCE GUIDANCE

In our IGD framework, influence guidance necessitates a surrogate model to be trained on the original dataset, collecting representative checkpoints for calculating guided loss. Here, we test the impact of influence guidance obtained over networks of different architectures, including ConvNet-6, ResNetAP-10, and ResNet18. We then train these networks on generated surrogate datasets from scratch and evaluate their cross-architecture performance. Table 4 demonstrates that datasets generated based on ConvNet-6 generally exhibit superior performance. In most cross-architecture evaluations involving ResNetAP-10 and ResNet-18, they even outperform datasets generated with

Table 4: Cross-architecture performance of DiT-IGD using different surrogate architectures to calculate influence guidance.

| Dataset | Surrogate | ConvNet-6 | | | ResNetAP-10 | | | ResNet-18 | | |
| --- | --- | --- | --- | --- | --- | --- | --- | --- | --- | --- |
| | | IPC10 | IPC50 | IPC100 | IPC10 | IPC50 | IPC100 | IPC10 | IPC50 | IPC100 |
| Nette | ConvNet-6 | 61.9±1.9 | **80.9±0.9** | **84.5±0.7** | **66.5±1.1** | 81.0±1.2 | **85.2±0.5** | **67.7±0.3** | 81.0±0.7 | 84.4±0.8 |
| | ResNetAP-10 | 58.9±0.4 | 79.5±0.8 | 83.7±0.4 | 66.2±0.8 | **82.3±0.2** | 84.4±0.8 | 66.7±0.6 | **82.3±0.9** | **85.4±0.4** |
| | ResNet18 | **62.2±0.3** | 78.5±0.9 | 80.1±0.3 | 63.3±1.6 | 79.5±0.8 | 82.1±0.5 | 63.1±0.3 | 80.3±0.7 | 83.3±1.4 |
| Woof | ConvNet-6 | **35.0±0.8** | 54.2±0.7 | **61.1±1.0** | **41.0±0.8** | **62.7±1.2** | **69.7±0.9** | **44.8±0.8** | 62.0±1.1 | **70.6±1.8** |
| | ResNetAP-10 | 33.8±1.0 | 53.5±0.3 | 60.0±0.4 | 39.6±0.4 | 61.5±0.8 | 68.8±0.5 | 43.6±0.5 | **65.5±0.7** | 69.3±0.4 |
| | ResNet18 | 34.3±0.8 | **54.3±0.8** | 61.0±1.8 | 39.5±1.1 | 61.0±1.4 | 68.7±0.7 | 43.8±1.4 | 62.9±1.0 | 69.5±0.4 |

Table 5: The ablation study of proposed influence guidance $\mathcal{G}_I$ and deviation guidance $\mathcal{G}_D$ tested with ResNet-18 on ImageNette .

| $\mathcal{G}_I$ | $\mathcal{G}_D$ | DiT-IGD | | Minimax-IGD | |
| --- | --- | --- | --- | --- | --- |
| | | IPC50 | IPC100 | IPC50 | IPC100 |
| ✗ | ✗ | 75.2±0.9 | 77.8±0.6 | 78.1±0.6 | 81.3±0.7 |
| ✓ | ✗ | 76.5±0.6 | 79.1±0.4 | 81.5±0.4 | 85.1±0.4 |
| ✗ | ✓ | 78.2±0.4 | 80.7±0.7 | 78.5±0.2 | 82.8±0.3 |
| ✓ | ✓ | **81.0±0.7** | **84.4±0.8** | **82.0±0.3** | **86.0±0.6** |

Table 6: Comparison of checkpoint selection strategies for Minimax-IGD: the gradient-similarity-based method versus regular interval selection, on ImageNette with ResNet-18.

| Threshold | # Checkpoints | Regular | Ours |
| --- | --- | --- | --- |
| 0.65 | 3 | 79.5±0.6 | 80.4±0.7 |
| 0.70 | 4 | 79.8±1.1 | **82.0±0.3** |
| 0.75 | 6 | 80.5±0.4 | 81.4±0.5 |
| 0.80 | 10 | 81.1±0.5 | 80.8±0.3 |

the test architecture. Additionally, due to fewer model parameters compared to the other two, the computational time required for influence loss calculations is reduced. Based on these observations, we choose to utilize ConvNet-6 as the surrogate in our formal implementation. However, we also note that the performance gap between datasets generated with different architectures is not significant. Particularly, datasets generated with ResNetAP-10 notably outperform ConvNet-6 in several tests against ResNet-18. These results further validate the robustness and generalization ability of our proposed IGD sampling framework.

## 4.4 ABLATION STUDY AND ANALYSIS

**Guidance component analysis.** Table 5 presents the performance achieved by independently applying influence guidance and deviation guidance to raw DiT and Minimax. The independent utilization of the two proposed guidance mechanisms still enhances the performance of both backbone methods. Specifically, in the case of raw DiT, the incorporation of deviation guidance yields results akin to those obtained with raw Minimax, primarily due to its ability to augment the diversity of generated data. Conversely, for Minimax, sole reliance on influence guidance markedly elevates its performance, achieving parity with the comprehensive framework. Despite Minimax's inherent focus on refining sample diversity through fine-tuning, additional gains can be attained through the integration of deviation guidance. Moreover, it is important to note that although influence guidance yields moderate improvements for raw DiT, the integration of deviation guidance results in significant enhancements. These observations substantiate our discourse regarding the critical role of data diversity in optimizing influence effectiveness. Conclusively, the synergy between influence guidance and deviation guidance complements each other, facilitating our guided sampling framework harmoniously in aligning with the training-enhancing objective.

**Early-stage guidance analysis.** We assess the practicability of our early-stage guidance strategy by comparing it with the entire guided sampling approach on ImageWoof, with variations in the influence guidance scaling factor $k$. Figure 2b demonstrates that applying the influence guidance throughout the entire generation stage with a large preset $k$ can significantly reduce influence loss. However, as illustrated by Figure 2c, when $k \geq 10$, despite a reduction in loss, validation accuracy notably drops, likely due to overfitting to the surrogate used for influence calculation. Moreover, this also leads to abnormal image generation shown in Figure 2a. In contrast, the early-stage guidance strategy allows strong guidance signals to steer the generation process effectively while mitigating the overfitting problem. Consequently, this strategy achieves superior performance in less generation time, thereby enhancing both the efficacy and efficiency of the process.

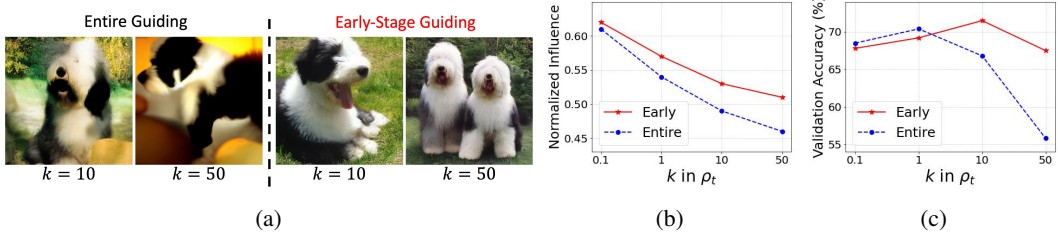

(a)                                (b)                                (c)

Figure 2: **(a)** Examples generated using entire and early-stage guidance with varying influence magnitude $k$ on ImageWoof; **(b)** Averaged normalized loss $\mathcal{G}_I$ of datasets generated with different values of $k$ and IPC=100; **(c)** Corresponding validation accuracies for varying $k$.

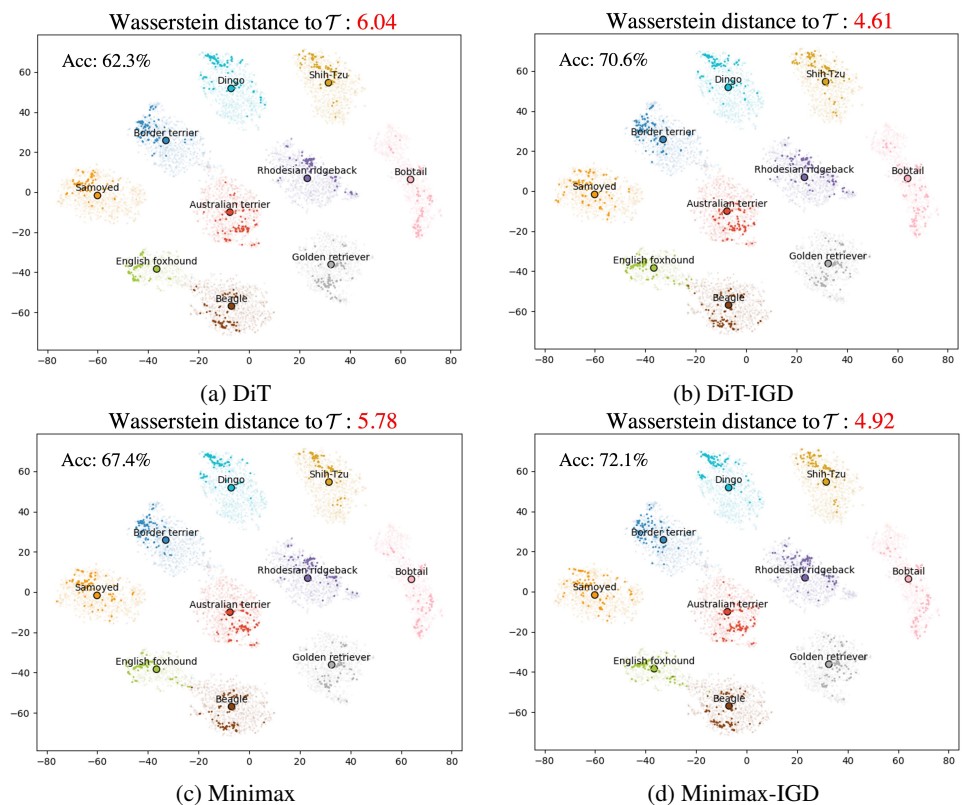

Figure 3: Visualization study for sample distributions of synthetic datasets (IPC=100) generated by four methodologies versus the original ImageWoof dataset. Smaller Wasserstein distances to the original dataset $\mathcal{T}$ signify closer alignment with the authentic distribution.

**Checkpoints selection strategy analysis.**     We assess the efficacy of the gradient-similarity-based checkpoint selection strategy proposed for computing the influence-guided loss (Equation (7)). A predetermined threshold is utilized to determine checkpoint selection based on the similarity of its averaged real gradient to the current reference, with an empirically identified suitable range set as $[0.6, 0.8]$. Thresholds beyond this range result in excessive checkpoint selection, leading to diminished efficiency, while overly small thresholds yield minimal selection. The baseline comparison involves the original trajectory influence's strategy, which saves checkpoints at fixed regular intervals. In Table 6, we contrast our strategy's results with various thresholds against the original regular strategy. To ensure fairness, an equal number of regularly collected checkpoints is used for guidance calculation at each threshold scenario. Comparative analysis reveals superior performance of our strategy over the regular approach. Notably, at a threshold of 0.7, our strategy with 4 checkpoints outperforms the results of 10 regularly selected checkpoints, demonstrating enhanced efficiency and efficacy. For a case study, checkpoint selection indexes of $\{0, 4, 11, 40\}$ are observed at a threshold of 0.7, compared

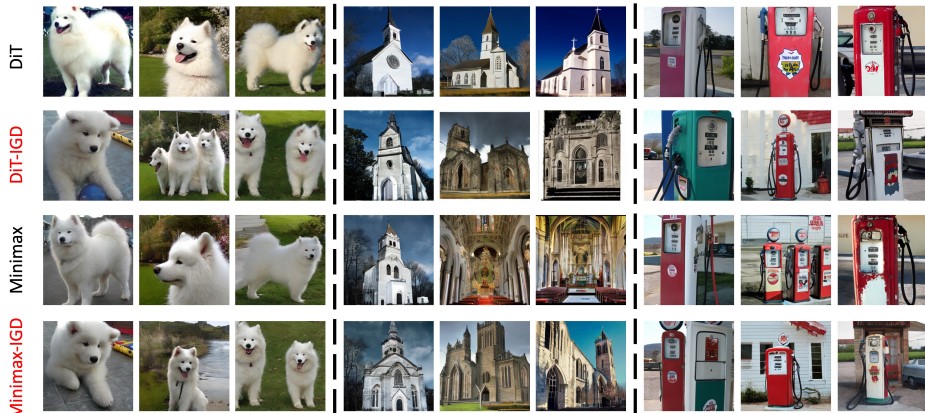

Figure 4: Comparison of image generation results from raw DiT, DiT-IGD, Minimax, and Minimax-IGD. Images in each column share the same random seed. Integrating IGD directly into the generation process produces high-quality data with varying semantic content and enhanced diversity compared to vanilla generation. Many instances exhibit robust consistency under the guidance of IGD.

to the regular indexes $\{0, 16, 32, 48\}$. This adaptive selection indicates better alignment with typical training dynamics, as more checkpoints are selected from the early stages of training.

### 4.5 Visualization Study on Generated Data

**Data distribution comparison.** To clearly investigate the effect of our guided sampling method on diffusion generation, Figure 3 shows t-SNE distribution comparisons among the full ImageWoof training dataset and data produced by two baseline methods, DiT and Minimax, as well as our two IGD-based approaches, each set at IPC=100. Additionally, we use the Wasserstein distance to quantitatively evaluate how well the distributions of the generated datasets align with the entire training dataset. Relative to the Minimax method, our IGD approach guides the diffusion process to achieve a closer match to the original training set's distribution, offering more comprehensive coverage and lower Wasserstein distances. Notably, Minimax-IGD surpasses DiT-IGD in performance, despite a higher Wasserstein distance from the original dataset. This finding lends partial support to our hypothesis that pinpointing a pivotal conditional distribution within the authentic distribution can be more beneficial than mere distribution alignment.

**Synthetic image comparison.** Figure 4 compares images generated by vanilla sampling of raw DiT and Minimax with those from guided sampling methods DiT-IGD and Minimax-IGD, using the same random seeds for each column. While baseline DiT generates high-quality images, they often share similar content, such as poses and structures. Minimax attempts to address the diversity issue in the generated data through fine-tuning DiT, but in many cases, the primary content or layout of the objects does not significantly change. In contrast, our method introduces additional signals in each guided generation step, achieving significant content variation and enhanced diversity without reducing quality. Furthermore, the guided signal from IGD is robust, producing similar content in both Minimax fine-tuned DiT and raw DiT in many cases.

## 5 Conclusion

In this work, we introduce a novel approach to dataset distillation by framing it as a guided diffusion generation problem. We correlate the general objective of dataset distillation with the trajectory influence function, designing an efficient influence-guided function for the diffusion sampling process. Additionally, we implement a deviation guidance function to ensure diversity and prevent training signal redundancy. These innovations enable us to create an efficient influence-guided diffusion sampling framework. Comprehensive experimental results illustrate that our method significantly improves the performance of diffusion models and demonstrate remarkable cross-architecture generalization ability.

ACKNOWLEDGEMENTS

Jiawei Du was supported by the A*STAR Career Development Fund (Grant No. C233312004). Yi Wang was supported in part by the Guangdong Basic and Applied Basic Research Foundation (Grant No. 2023B1515120058). Wei Wang was supported by the Guangdong Provincial Key Laboratory of Integrated Communication, Sensing, and Computation for Ubiquitous Internet of Things (Grant No. 2023B1212010007), the Guangzhou Municipal Science and Technology Project (Grant Nos. 2023A03J0003, 2023A03J0013, and 2024A03J0621), and the Institute of Education Innovation and Practice Project (Grant Nos. G01RF000012 and G01RF000017).

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

# A    APPENDIX

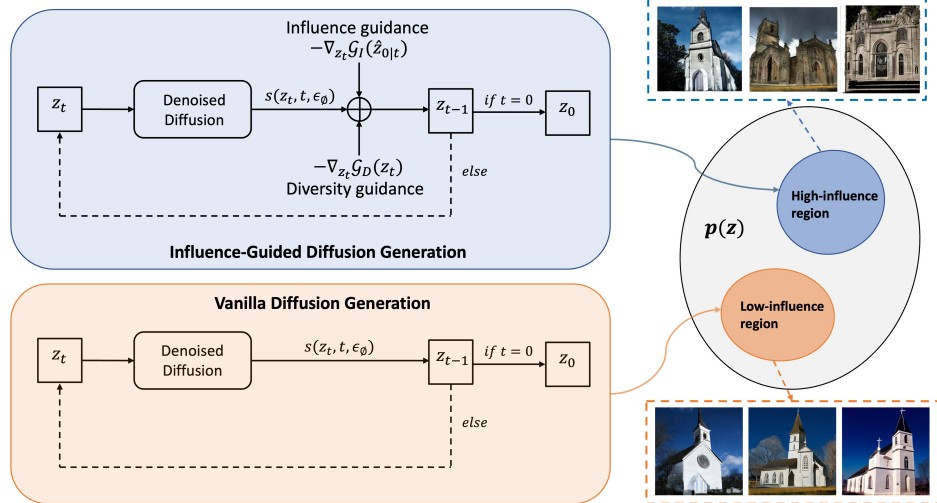

Figure 5: An intuitive comparison between our influence-guided diffusion generation and the vanilla diffusion generation frameworks.

## A.1    RELATED WORK

**Dataset Distillation.**    Current dataset distillation methods can be categorized into meta-learning, data-matching, and model-inversion approaches. Meta-learning methods (Nguyen et al., 2021; Zhou et al., 2022; Loo et al., 2023) tackle dataset distillation as a nested optimization problem, aiming to minimize generalization errors on original data caused by models trained on distilled data. Data-matching methods involve synthesizing data to replicate specific behaviours from the original dataset, such as latent distributions (Zhao & Bilen, 2021b; Wang et al., 2022a), gradients (Zhao et al., 2021; Zhao & Bilen, 2021a; Kim et al., 2022), training trajectories (Cazenavette et al., 2022; Du et al., 2023; Cui et al., 2023) and surrogate predictions (Chen et al., 2023). Model-inversion methods (Yin et al., 2024; Shao et al., 2023) are established on data-free knowledge distillation (DFKD) techniques with specific batch normalization statistic alignment. Additionally, recent research (Gu et al., 2024) has integrated diffusion models into dataset distillation alongside a fine-tuning scheme, perpendicular to our training-free, sampling-oriented approach.

**Guided-Diffusion Sampling.**    Works in this category employ a pre-trained diffusion model as a foundation but modify the sampling method to guide generation with feedback from the guidance function (Kawar et al., 2022; Chung et al., 2022; Graikos et al., 2022). Early work employed classifiers as guidance, adjusting gradients during sampling (Dhariwal & Nichol, 2021). However, classifiers for noisy images are domain-specific and often unavailable. (Wang et al., 2022b) introduced linear operator-based external guidance, generating images in the null space of these operators, though extending to non-linear functions is challenging. Several recent works (Gopalakrishnan Nair et al., 2023; Yu et al., 2023; Bansal et al., 2023) explored general guidance functions, modifying the sampling process with gradients of the guidance function on expected denoised images. However, these methods rely on existing metric functions that can concretely measure specific requirements. In contrast, our contribution lies in guiding the model to generate data that meets abstract, training-enchaining criteria.

## A.2    LIMITATIONS AND FUTURE WORK

The main limitation of our method is the additional time incurred by guidance calculations during the diffusion sampling process. Despite efforts to improve efficiency, our sampling framework takes 5 to 6× longer than the vanilla method. For example, raw DDIM generates a $256 \times 256$ image in ~1.5 seconds, our method takes ~8.2 seconds on a RTX 4090 GPU. This is particularly challenging

for distilling extensive datasets in resource-constrained scenarios. Consequently, improving the generation efficiency of guided diffusion sampling method will be a key focus of our future research.

### A.3    GRADIENT-SIMILARITY-BASED CHECKPOINT SELECTION ALGORITHM

In Algorithm 2, we present a detailed implementation of the gradient-similarity-based checkpoint selection algorithm introduced in Section 3.2. This algorithm is designed to select representative checkpoints for calculating the influence guidance $\mathcal{G}_I$. The core intuition behind this algorithm is that if the gradients at a checkpoint closely resemble those at the previous one, the previous checkpoint can effectively represent the current one.

**Complexity analysis.**    The computational overhead primarily stems from calculating the averaged gradient $g_t$ w.r.t the model parameters $\theta$ at each of the $E$ checkpoints collected during training. When using the same cross-entropy loss as in model training, due to its **additive nature**, the computational complexity of calculating $g_t$ at a given checkpoint $\theta_t$ is equivalent to the complexity of one epoch of gradient descent, approximately $O(|\theta| \cdot B \cdot \frac{N}{B} \cdot d)$, where $B$ is the batch size, $N$ is the number of data instances, and $d$ is the data dimension. Essentially, without any optimization, the complexity of this algorithm is similar to training a model parameterized by $\theta$ for $E$ epochs. In practice, instead of loading the entire dataset into the dataloader to compute the average gradient $\bar{\nabla}_\theta \ell_c$ for each class, we first load all images from a class folder into CPU memory and slice them into GPU memory for gradient computation and accumulation. Empirically, this approach further reduces the runtime of the filtering algorithm. Additionally, the cross-architecture evaluation discussed in Section 4.3 and Table 4 demonstrates that using models with simpler architectures (e.g., ConvNet) as surrogates can provide more effective influence guidance, further reducing the time overhead for selecting representative checkpoints.

---

**Algorithm 2:** Filtering Algorithm for Influence Guidance

**Input:** Original dataset $\mathcal{T}$, Initial checkpoint $\theta_0^{\mathcal{T}}$, Threshold $\delta$
**Output:** Retained checkpoints list $\mathcal{R}$

1  **Initialize:** $\mathcal{R} \leftarrow \theta_0^{\mathcal{T}}$;
2  Compute $\mathbb{E}_c[\bar{\nabla}_\theta \ell_c(\mathcal{X}_c; \theta_0^{\mathcal{T}})]$ as reference gradient $g_{\text{ref}}$;
3  **for** $t = 1$ **to** $E$ **do**
4  $\quad$ Compute averaged gradient $g_t = \mathbb{E}_c[\bar{\nabla}_\theta \ell_c(\mathcal{X}_c; \theta_t^{\mathcal{T}})]$;
5  $\quad$ Calculate cosine similarity $s = \frac{g_t \cdot g_{\text{ref}}}{\|g_t\| \|g_{\text{ref}}\|}$;
6  $\quad$ **if** $s < \delta$ **then**
7  $\quad\quad$ $\mathcal{R} \leftarrow \mathcal{R} \cup \{\theta_t^{\mathcal{T}}\}$;
8  $\quad\quad$ Update reference gradient $g_{\text{ref}} = g_t$;
9  **return** $\mathcal{R}$;

---

### A.4    ADDITIONAL PERFORMANCE EVALUATION ON FOOD-101 DATASET

We evaluate the performance of our IGD methods on Food-101 (Bossard et al., 2014) dataset to provide further test on distilling other large, high-resolution datasets. Food-101 is a challenging dataset that includes 101 food categories, totaling 101,000 images, with each category containing 250 manually reviewed test images and 750 training images. All images are scaled to a maximum side length of 256 pixels. Results detailed in Table 7 show that our IGD methods achieve superior performances over all IPC scenarios. Furthermore, applying our method to baseline methods, including DiT and Minimax, results in noticeable performance enhancements, with average improvements of 3.8% and 3.5%, respectively. In contrast, the Minimax method yields only a marginal average improvement of 0.8% to DiT. These findings align with evaluations conducted on ImageNet, indicating robust scalability across large, high-resolution datasets.

### A.5    ADDITIONAL PERFORMANCE EVALUATION ON CIFAR DATASETS

The results presented in Tables 1 and 2 demonstrate the outstanding performance of our method in distilling targeted high-resolution datasets. In this section, we further investigate the generalizability of our method to smaller datasets. Specifically, we compare the performance of our framework with

Table 7: **Food-101:** performance comparison with state-of-the-art pixel-level distillation methods, pretrained DiT and Minimax-tuned DiT models with vanilla generation. The results are obtained on ResNetAP-10 at different IPCs.

| IPC | Random | DM | DiT | DiT-IGD | Minimax | Minimax-IGD | Full |
|-----|--------|----|----|----------------|---------|--------------------|------|
| 10 | 16.2±0.5 | 18.5±0.8 | 23.9±1.0 | 27.2±0.9 | 24.8±0.9 | **28.3±0.9** | |
| 50 | 36.9±0.3 | 37.8±0.4 | 40.8±0.7 | **45.2±0.7** | 41.6±1.0 | 44.8±0.7 | 78.6±0.4 |
| 100 | 46.8±0.3 | 44.8±0.3 | 45.9±0.5 | 49.7±0.3 | 46.5±0.5 | **50.3±0.6** | |

Table 8: **CIFAR-10 & CIFAR-100:** Perfomrance comparison with two low-resolution-orientied methods DM and DATM and two high-resolution-oriented methods SRe$^2$L and RDED.

| | CIFAR-10 | | | CIFAR-100 | | |
|-----|------|------|------|------|------|------|
| IPC | 50 | 500 | 1000 | 10 | 50 | 100 |
| Ratio (%) | 1.0 | 10.0 | 20.0 | 2.0 | 10.0 | 20.0 |
| DM | 63.1±0.4 | 74.3±2 | 79.2±0.2 | 29.7±0.3 | 43.6±0.4 | 47.1±0.4 |
| DATM | 76.1±0.3 | 83.5±0.2 | 85.5±0.4 | 47.2±0.4 | 55.0±0.2 | 57.5±0.2 |
| SRe$^2$L | 43.2±0.3 | 55.3±0.4 | 57.1±0.4 | 24.5±0.4 | 45.2±0.3 | 46.6±0.5 |
| RDED | 68.4±0.2 | 78.1±0.4 | 79.8±0.4 | 46.4±0.3 | 51.5±0.3 | 52.6±0.4 |
| DiT-IGD | 66.8±0.5 | 82.6±0.6 | 84.6±0.5 | 45.8±0.5 | 53.9±0.6 | 55.9±0.4 |

two state-of-the-art high-resolution-oriented methods, SRe$^2$L Yin et al. (2024) and RDED Sun et al. (2024), as well as two low-resolution-oriented methods, DM Zhao & Bilen (2021b) and DATM Guo et al. (2024), on CIFAR-10 and CIFAR-100. Notably, DATM, a recent strong baseline, has been shown to achieve lossless distillation on small-scale datasets such as CIFAR. Table 8 presents the comparison results on ConvNet under varying IPC values. Our experimental findings indicate that our method consistently outperforms both SRe$^2$L and RDED across most scenarios. Remarkably, our approach achieves nearly lossless performance, comparable to that of DATM, even at a 20% compression ratio. These results, combined with the exceptional performance observed on larger datasets like ImageNet, suggest that our method is a versatile and unified solution that excels in both low-resolution and high-resolution settings.

## A.6 COMPARISON WITH OTHER DIFFUSION-BASED METHODS ON IMAGENET-1K

We compare our method with two other synthetic dataset generation methods, namely TDSDM (Yuan et al., 2023) and D$^4$M (Su et al., 2024), which leverage pre-trained diffusion models. Although TDSDM was not initially designed for dataset distillation tasks, its goal is to enhance the training efficacy of synthetic data generated by diffusion models. Along with our baseline method, Minimax, these three approaches utilize distribution-matching-like objectives to fine-tune diffusion models and improve the performance of synthetic data in training. The results presented in Table 9 are evaluated on ImageNet-1K under the evaluation protocol of RDED. Our findings demonstrate that our guided-diffusion method consistently outperforms the others, reinforcing the importance of introducing informative guidance when applying diffusion models in dataset distillation.

Table 9: Performance comparison over ResNet-18 with state-of-the-art diffusion-finetuning methods

| Dataset | IPC | TDSDM | D$^4$M | Minimax | DiT-IGD | Minimax-IGD |
|---------|-----|-------|-------|---------|---------|-------------|
| ImageNet-1K | 10 | 44.5±0.4 | 27.9±0.7 | 44.3±0.5 | 45.5±0.5 | **46.2±0.6** |
| | 50 | 59.4±0.3 | 55.2±0.3 | 58.6±0.3 | 59.8±0.3 | **60.3±0.4** |

A.7 Evaluating the Robustness of Influence-Guided Diffusion with DPM Solver

In the practical implementation of our IGD methods, we propose using DDIM with 50 denoising steps. To assess the generalizability of our proposed Influence Guidance and Deviation Guidance across different diffusion solvers, we additionally explore the use of the DPM solver Lu et al. (2022) as a replacement for the DDIM solver in outputting $s(z_t, t, \epsilon_\phi)$ in Equation (9). The DPM solver is a fast, high-order solver for diffusion ODEs with a convergence order guarantee. We employ the second-order DPM solver with 20 denoising steps by default. Accordingly, we adjust the guided range for diffusion guidance to [12, 18]. This adjustment results in a significant 50% reduction in average sampling time, from 8.2 seconds to 4.3 seconds on an RTX 4090. Table 10 compares the average performance of DDIM with 50 steps and the DPM solver with 20 steps for distilling ImageNette and ImageWoof with IPC=50. The results indicate that using the DPM solver with fewer denoising steps does not lead to a significant degradation in performance, and in some scenarios, it even yields slight improvements. This further validates the robustness of our influence-guided sampling method.

Table 10: Performance of ResNet-18 using the DDIM-50 and DPM-20 solvers for diffusion generation.

|  | Solver | DiT | DiT-IGD | Minimax | Minimax-IGD |
|---|---|---|---|---|---|
| ImageNette | DDIM-50 | 75.2±0.9 | 81.0±0.7 | 78.1±0.6 | **82.0±0.3** |
|  | DPM-20 | 73.5±0.8 | **82.0±0.5** | 76.6±0.4 | 80.6±0.5 |
| ImageWoof | DDIM-50 | 57.4±0.7 | 62.0±1.1 | 60.5±0.5 | **65.4±1.8** |
|  | DPM-20 | 57.8±0.6 | 64.0±1.0 | 60.3±0.6 | **64.5±1.3** |

A.8 Distribution Diversity and Coverage Analysis

In Figure 3, we provided t-SNE visualizations comparing the distributions of data generated by two baseline methods (DiT and Minimax) and our IGD-based methods with IPC=100. The figure shows that integrating IGD enhances diversity and alignment with the original dataset, supported by lower Wasserstein distances to the original dataset. In this section, we further compare the FID scores and coverage of surrogate datasets (IPC=100) generated for ImageWoof by different methods. Here, coverage metric was assessed based on whether each original data point had a nearest neighbor in the surrogate dataset within a given threshold (e.g., 300 in the Inception V3 latent space). For fairness, we excluded data selected by the Random method from the original dataset during coverage calculation. From the results shown in Table 11 also demonstrate a clear diversity improment achieved by our IGD method to vanilla DiT. However we also observed that although the randomly selected dataset has the lowest FID and highest coverage, its performance was the worst. Similarly, while Minimax-IGD has worse FID and coverage than DiT-IGD, it performed better. These findings suggest that our diversity-constraint influence-guided objective is a more effective measure for DD than relying solely on distribution alignment.

Table 11: FID and distribution coverage comparison among different methods.

| Metrics | DiT | DiT-IGD | Minimax | Minimax-IGD | Random |
|---|---|---|---|---|---|
| FID | 81.1 | 75.9 | 80.1 | 76.4 | **54.1** |
| Coverage (%) | 65.4 | 68.1 | 66.5 | 67.2 | **72.1** |
| Accuracy (%) | 62.3 | 70.6 | 67.4 | **72.1** | 63.6 |

A.9 Parameter Analysis

In our IGD sampling framework, two critical hyper-parameters are $k$, which controls the magnitude of influence guidance, and $\gamma_t$, which controls the magnitude of deviation guidance. In Figure 6, we examine the impact of these scaling factors on DiT-IGD and Minimax-IGD using the ImageNette dataset as an instance. For DiT-IGD, variations in both $k$ and $\gamma_t$ significantly influence performance.

Increasing the values of these parameters enhances performance, highlighting the importance of influence and dataset diversity for model training. However, setting $k$ too high results in a notable performance drop. As discussed in Section 3.2, this is likely due to excessive overfitting to the surrogate data with distorted content. In contrast, for Minimax-IGD, increasing $\gamma_t$ contributes marginally to performance improvement. This is because Minimax-IGD inherently focuses on increasing diversity as a core aspect of its fine-tuning-based scheme. However, increasing influence guidance by enlarging $k$ significantly improves its results. Despite this improvement, a similar performance drop is observed when $k$ becomes excessively large. These findings underscore the necessity of carefully tuning $k$ and $\gamma_t$ to optimize the effectiveness of our IGD sampling framework, ensuring balanced influence and diversity without overfitting.

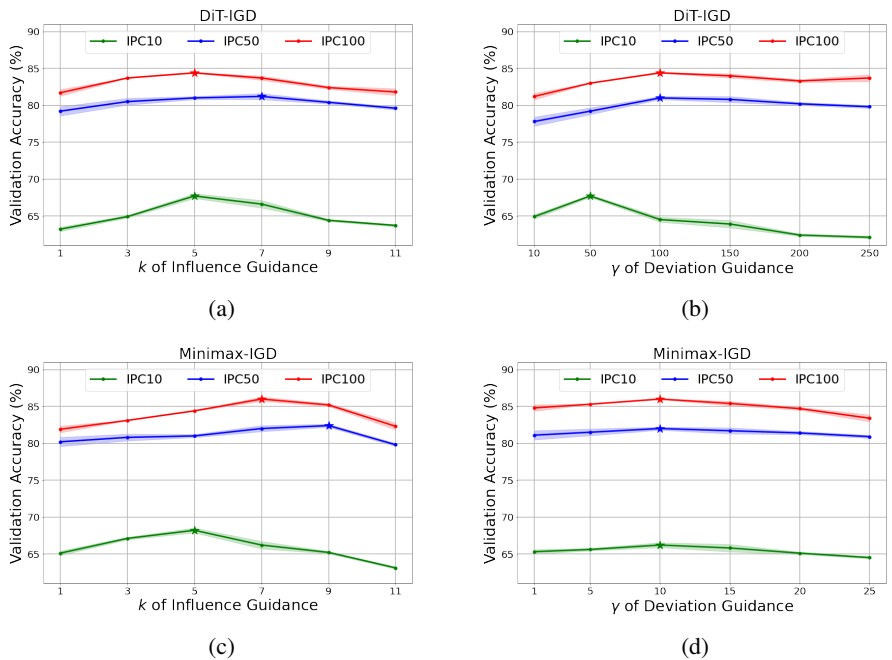

Figure 6: Hyper-parameter analysis on (a) & (c) the scaling factor $k$ of influence guidance, and (b) & (d) the scaling factor $\gamma_t$ of deviation guidance for DiT-IGD and Minimax-IGD.

### A.10    Hyperparameter Setup and Guidelines

In Table 12, we provide a detailed hyperparameter configuration for $k$ and $\gamma_t$ in Equation (7) to replicate the results obtained across ImageNette, ImageWoof, and ImageNet-1K datasets. Despite incorporating an adaptive scaling factor based on the ratio between the denoised signal magnitude from diffusion and the guided signal from the influence guidance $\mathcal{G}_I$, manual specification of the scale factor $k$ remains essential to forestall unexpected overfitting resulting from the influence guidance. Drawing from insights gleaned from our ablation study, as illustrated in Figure 6, we recommend setting the value range of $k$ within $[1, 50]$ for scaling our method in distillation tasks involving other ImageNet subsets. Similarly, we suggest a grid-search range for the scaling factor $\gamma_t$ of the deviation guidance as $[10, 200]$. Particularly for scenarios with small IPC, we advocate for starting from a relatively smaller value of $k$ to hold the representatives of generated data.

### A.11    More visualization comparison of synthetic data.

Here, we provide an additional visual comparison between images generated by two backbone models with vanilla DDIM sampling: the raw DiT and the Minimax-tuned DiT, and with our IGD-sampling framework: DiT-IGD and Minimax IGD. All synthetic data were generated for the ImageWoof and ImageNette datasets.

Table 12: Detailed setup of hyperparameters $k$ and $\gamma_t$ in Equation (7) for reproducing the results reported in Table 1 & 2.

| | | DIT-IGD | | | Minimax-IGD | | |
|---|---|---|---|---|---|---|---|
| | Parameter | IPC10 | IPC50 | IPC100 | IPC10 | IPC50 | IPC100 |
| Nette | $k$ | 5 | 5 | 5 | 15 | 15 | 15 |
| | $\gamma_t$ | 50 | 120 | 120 | 10 | 10 | 10 |
| woof | $k$ | 5 | 5 | 5 | 10 | 10 | 10 |
| | $\gamma_t$ | 50 | 120 | 120 | 50 | 100 | 100 |
| 1K | $k$ | 5 | 5 | 5 | 10 | 10 | 10 |
| | $\gamma_t$ | 120 | 120 | 120 | 100 | 100 | 100 |

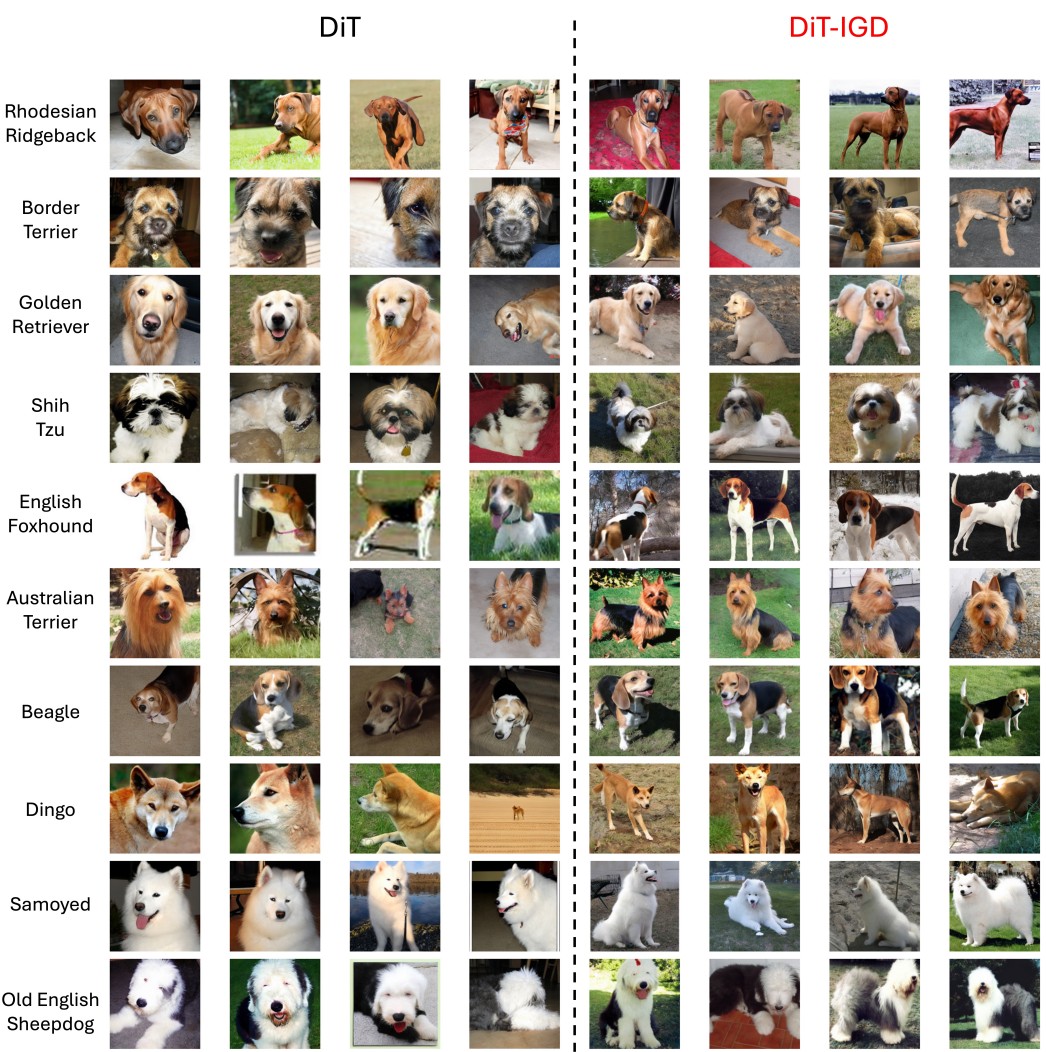

Figure 7: Visualizaiton comparison between raw DiT and DiT-IGD on ImageWoof.

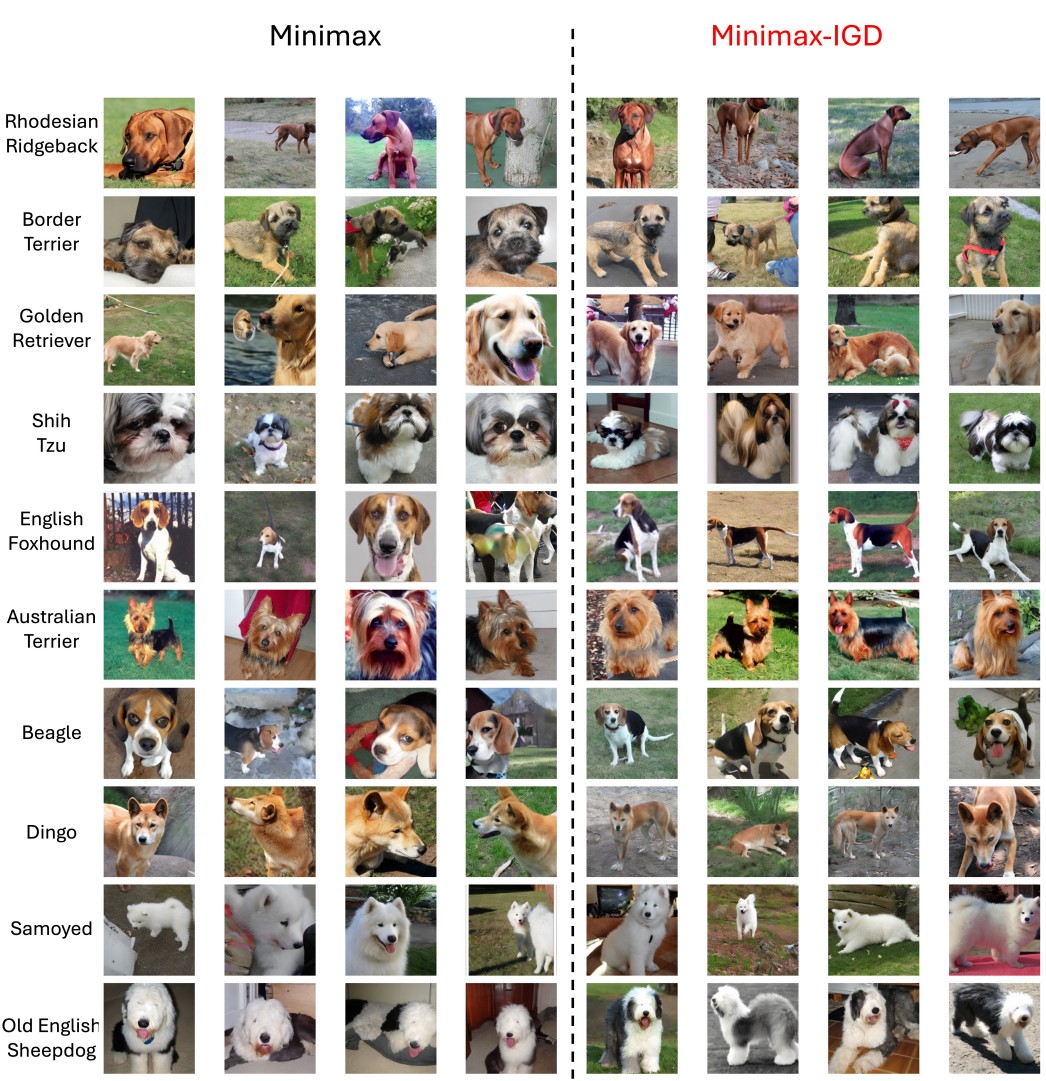

Figure 8: Visualizaiton comparison between Minimax and Minimax-IGD on ImageWoof.

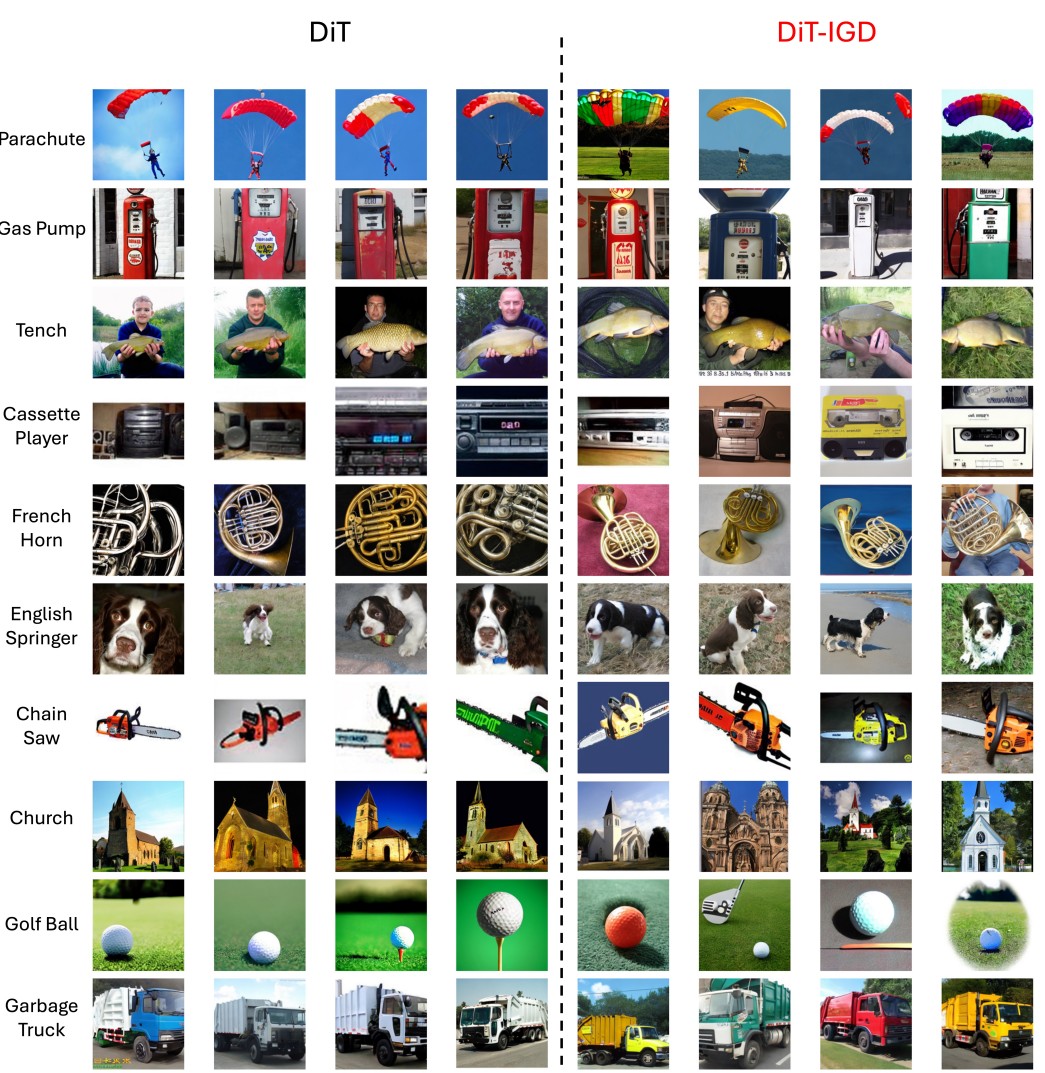

Figure 9: Visualizaiton comparison between raw DiT and DiT-IGD on ImageNette.

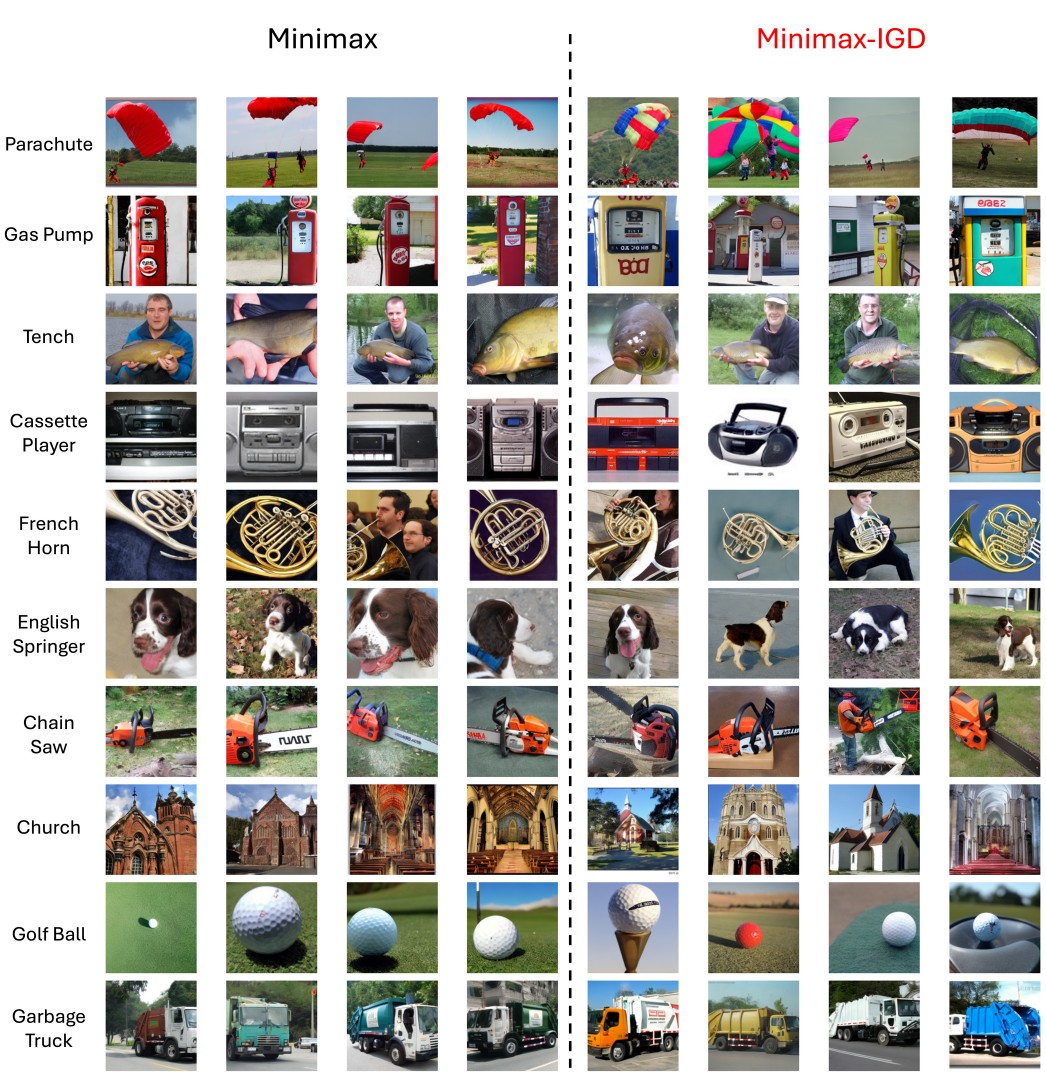

Figure 10: Visualizaiton comparison between Minimax and Minimax-IGD on ImageNette.

