# OpenReview forum: "Influence-Guided Diffusion for Dataset Distillation"
_ICLR.cc/2025/Conference — ICLR 2025 Poster_

### Official Review · Reviewer_Qttw · 2024-10-29

**Soundness:** 3
**Presentation:** 4
**Contribution:** 3
**Rating:** 8
**Confidence:** 4

**Summary:**

This paper proposed a guided diffusion generation method for dataset distillation problem. The trajectory influence and deviation guidance are introduced to the vanilla diffusion process for generating synthetic samples as efficient training data. The results on ImageNet and its subsets demonstrates the improvements over the baselines.

**Strengths:**

Strengths:

1.	The paper is easy to follow. The motivation of importance guided synthesis is clear and method is well presented.

2.	The new idea is reasonable and neat that bridge the diffusion-based generative models and importance-based sample selection.

3.	The results are promising. The performance improvements on challenging datasets are remarkable. Extensive ablation studies, cross-architecture validation and visualization are implemented.

**Weaknesses:**

Weaknesses:

1.	There are several hyper-parameters in the algorithm, such as influence factor, deviation factor, scale, guided range etc. The sensitiveness of performance on these hyper-parameters should be studied. How do authors search the hyper-parameters?

2.	I have a concern whether the new method cause the collapse of the distribution of the generated data, though with the deviation guidance.

3.	Since DiT and VAE pre-trained on huge dataset are utilized for generating training samples on small datasets. It naturally brings advantages over traditional dataset distillation methods. Hence, more recent methods that also use pre-trained diffusion models should be compared with.

4.	There are also some similar works that improve the efficiency of diffusion-model generated training samples, such as [1], which should also be discussed in the paper.

[1] Real-Fake: Effective Training Data Synthesis Through Distribution Matching, ICLR 2024.

**Questions:**

Please address the above weaknesses.

---

> ### Author Response · Authors · 2024-11-20
> **Response to Reviewer Qttw (1/2)**
>
> Thank you very much for your positive feedback! We greatly appreciate your insightful question regarding the distribution  of the generated data, which led us to discover new insights on applying our guided-diffusion method for dataset distillation. Below, we address your questions in sequence.
>
> ---
> **Q1(@Weakness1): The performance sensitivity to hyper-parameters should be examined, including how to conduct the hyper-parameter search.**
>
> **A1:** Thank you for your valuable suggestion. **We have already analyzed the impact of the influence factor $k$ and deviation factor $\gamma$ in Appendix A.5**, titled "Parameter Analysis", and in **Figure 5** of the original paper. The optimal values for these factors were determined through a grid search. As shown in Figure 5, our DiT-IGD method is relatively sensitive to changes in both factors, which supports our motivation for adding a diversity constraint to the influence guidance. In contrast, since the baseline Minimax has already strengthened diversity through fine-tuning, our Minimax-IGD primarily relies on adjusting the influence factor $k$.
>
> As for the guided range parameters, we drew on insights from previous guided diffusion work (line 266) to determine the them. This led us to search for the optimal range within steps $[25,50]$. For the length of the guided range, we tested values of 10, 15, and 20 steps, adjusting with a stride of 5 within $[25,50]$. Setting the length to 15 produced the best and most stable performance among various groups of $k$ and $\gamma$. Ultimately, we discovered that a guided range of $[30,45]$ achieved the best results. We provide a comparison for DiT-IGD on ImageWoof with IPC=100 as a reference.
>
> |     Range      | [25, 40] | [30, 45] | [35, 50] |
> |:--------------:|:--------:|:--------:|:--------:|
> | Acc (%)    |   69.9   |  **70.6**   |   67.2   |
>
> ---
> **Q2(@Weakness2): Whether the new method causes the collapse of the distribution of the generated data, though with the deviation guidance.**
>
> **A2:** In Section 4.5 and **Figure 3**, we provided t-SNE visualizations comparing the distributions of data generated by two baseline methods (DiT and Minimax) and our IGD-based methods with IPC=100. The figure shows that integrating **IGD enhances diversity and alignment** with the original dataset, supported by lower Wasserstein distances to the original dataset.
>
> Our further experiments reveal that "**focusing solely on diversity or alignment with the original dataset is insufficient for optimal effectiveness in DD scenarios**". We compare the **FID scores** and **coverage** of surrogate datasets (IPC=100) generated for ImageWoof by different methods,
>
> |            Metric           |   DiT   | DiT-IGD | Minimax  | Minimax-IGD | Random  |
> |:---------------------------:|:-------:|:-------:|:--------:|:-----------:|:-------:|
> | **FID**                     |  81.1   |  75.9   |   80.1   |     76.4    | **54.1**|
> | **Coverage (%)**            |  65.4   |  68.1   |   66.5   |     67.2    | **72.3**|
> | **Accuracy (%)**            |  62.3   |  70.6   |   67.4   |   **72.1**  |  63.6   |
>
> Coverage was assessed based on whether each original data point had a nearest neighbor in the surrogate dataset within a given threshold (e.g., 300 in the Inception V3 latent space). For fairness, we excluded data selected by the Random method from the original dataset during coverage calculation.
>
> From the results, although **the randomly selected dataset has the lowest FID and highest coverage, its performance was the worst**. Similarly, while Minimax-IGD had worse FID and coverage than DiT-IGD, it performed better. These findings suggest that our diversity-constraint influence-guided objective is a more effective measure for DD than relying solely on distribution alignment.

---

> ### Author Response · Authors · 2024-11-20
> **Response to Reviewer Qttw (2/2)**
>
> **Q3(@Weakness3&4): The paper should compare with other recent methods using pre-trained diffusion models.**
>
> **A3:** Thank you for the valuable suggestion and for introducing the insightful related work [1]. Although [1] was not originally proposed for the DD setting, it aims to improve the training efficacy of synthetic data generated by diffusion models. We also identified a related work, D$^4$M [2]. Together with our baseline method Minimax, **these three works propose using distribution-matching-like objectives to fine-tune diffusion models**. Here, we follow RDED’s evaluation protocol commonly used in dataset distillation (involving predicted soft labels) to compare our guided-diffusion methods with these three approaches on ImageNet-1K.
> | IPC | [1]  |   D$^4$M   |  Minimax  | DiT-IGD | Minimax-IGD |
> |:---:|:----:|:-------:|:---------:|:-------:|:-----------:|
> |  10 | 44.5 |   27.9  |   44.3    |   45.5  |   **46.2**  |
> |  50 | 59.4 |   55.2  |   58.6    |   59.8  |   **60.3**  |
>
> The results demonstrate that **our guided-diffusion methods continue to achieve superior performance**, supporting our claim that **introducing informative guidance is essential for applying diffusion models in dataset distillation**. We also observe that among the three distribution-matching-like fine-tuning methods, [1] achieves the best performance. We will include these evaluations in the later revised version of the paper and plan to explore integrating [1] with our influence-guided generation in future work.
>
> [1] Jianhao Yuan, Jie Zhang, Shuyang Sun, Philip Torr, Bo Zhao. Real-Fake: Effective Training Data Synthesis Through Distribution Matching. ICLR 2024
>
> [2] Duo Su, Junjie Hou, Weizhi Gao, Yingjie Tian, Bowen Tang. D4M: Dataset Distillation via Disentangled Diffusion Model. CVPR 2024

---

> ### Author Response · Authors · 2024-11-25
>
> Dear Reviewer Qttw,
>
> We are truly grateful for your constructive feedback and recognition of our work! Below is a summary of our responses to your questions:
>
> - A kind reminder of hyper-parameter analysis in Appendix A.5 and a detailed searching process of the guided range are introduced.
> - Additional experiments regarding data diversity and distribution coverage are discussed.
> - Comparisons with recent approaches using pre-trained diffusion models are involved, demonstrating a continued superior performance of our method.
>
> If there are any additional questions or thoughts, we are welcome to further discussion!
>
> Best Regards

---

> > ### Comment · Reviewer_Qttw · 2024-11-25
> > **Reviewer Response**
> >
> > Thanks for the detailed reponse and the new experiments. I believe the paper can be further enhanced after the revision.

---

> > ### Author Response · Authors · 2024-11-25
> >
> > Dear Reviewer Qttw,
> >
> > Thank you very much for your positive feedback on our responses and additional content! Reported comparisons with recent approaches using pre-trained diffusion models will be included in the revised version of the paper.
> >
> > We greatly appreciate your positive support of our submission!
> >
> > Best regards.

---

### Official Review · Reviewer_hMXA · 2024-11-02

**Soundness:** 3
**Presentation:** 3
**Contribution:** 3
**Rating:** 6
**Confidence:** 4

**Summary:**

This paper works on dataset distillation by generating the distilled dataset using diffusion models guided by an influence function. In the implementation, two guidance terms are used. One is to increase the similarity between the gradient using the generated sample and the average gradient using the original training samples. The other is to decrease the similarity between generated samples. Experiments are conducted on ImageNette and ImageWoof.

**Strengths:**

1. The idea of using guided diffusion to generate samples for distillation is an interesting application of diffusion models.

2. The proposed two guidance terms, i.e., increasing the gradient similarity and sample diversity, are well-motivated, simple, and intuitive.

3. The paper is well written and presented in general.

4. In the experiments, the proposed method achieves better performance and shows effectiveness. Comprehensive ablation studies and analyses are provided.

**Weaknesses:**

1. There are a lot of writing issues in the math part:
- It is unclear how the derivation is transferred from stepwise (Eq4) to epochwise (Eq5).
- C duplicated defined on L096 and L110.
- In Sec 2.2, some z is bold, and some are not.
- In L132, D is not clearly defined.
- In Eq5, theta_e and theta_E are not clearly defined.

2. The proposed method seems to have a high computational cost. Both computing and storage costs are high for the gradient calculation in L294. The similarity calculation with respect to all generated samples is also high in Eq8.  The computing and storage costs should be clearly provided and analysed in all the experiment sections.

3. I doubt the statement this method is training-free. I agree that it is training-free as commonly understood in the diffusion community. But there are still a lot of training efforts here. It is only training-free given all the checkpoints, stored gradients, and pre-trained diffusion models. I would suggest revising this statement.

**Questions:**

Please refer to W1 and W2.

---

> ### Author Response · Authors · 2024-11-20
> **Response to Reviewer hMXA (1/1)**
>
> Thank you for your positive feedbacks. We address your questions in the following responses.
>
> ---
> **Q1(@Weakness1): There are writing issues in the math part.**
>
> **A1:** Here are responses to the issues you raised:
>
> I. *It is unclear how the derivation is transferred from stepwise (Eq4) to epochwise (Eq5).*
>
> Answer: Based on the original formulation of the trajectory influece (TracIn), in Eq. (4), the change in loss between subsequent parameter updates is approximated using a first-order Taylor expansion, measuring the influence stepwise for each parameter update, representing the contribution of a training pair $(x, y)$ to the validation loss on $(x', y')$. By analogy to the fundamental theorem of calculus, the sum of the influences of all training examples on a fixed test point $(x', y')$ is equivalent to the total reduction in loss on $(x, y)$ during training. Thus, for a particular training point $(x, y)$, **the idealized influence can be approximated by summing the influence over all iterations where $(x, y)$ was used to update the parameters**. In practical SGD-based training, since each training point is used once per epoch, TracIn approximates the influence of $(x, y)$ on $(x', y')$ by summing influences collected at checkpoints across epochs.
>
> II. *C duplicated defined on L096 and L110, some z are bold and some are not in Sec 2.2 and D is not clearly defined in L132.*
>
> Answer: Thank you very much for your attention to detail and for pointing out these issues. First, we update the notation for the conditional distribution from $p(x|C)$ to $p(x|\textit{condition})$ to avoid confusion. Second, we correct the missing bold formatting. Third, we clearly indicate that $D(.)$ represents the decoder function and that $u_i=D(z_i)$. **All these revisions are updated in the current revised version of our paper**.
>
> ---
> **Q2 (@Weakness2): The proposed method seems to have high computational and storage costs, particularly for gradient (L294) and similarity calculations (Eq. 8). Detailed costs should be provided.**
>
> **A2:** Thank you for the suggestion. We repectfully clarify that our method does not require high computational resources to generate surrogate datasets. All experiments conducted on ImageWoof, ImageNette, and ImageNet-1K can be completed using a single RTX 4090 with **nearly 16 GB peak memory usage**. Regarding storage costs, peak usage occurs before applying our proposed gradient-similarity-based checkpoint selection algorithm to retain representative checkpoints. This involves storing $E+1$ model checkpoint parameters, but after selection, peak storage is reduced to approximately $2K$ checkpoints. With ConvNet-6 as the surrogate in our default implementation, **this cost nearly 295 MB for storing overall $51$ checkpoints' parameters** and **nearly 24 MB for storing $4$ selective checkpoints** and **nearly 24 MB for storing corresponding averaged gradients**.  Below, we provide a detailed analysis.
>
> First, as outlined in L290-L295, we obtain $E$ checkpoints trained on $\mathcal{T}$ and retain $K$ representative checkpoints for subsequent influence guidance calculation (Eq. 7). As shown in Table 6, our checkpoint selection algorithm allows for only 4 or 5 representative checkpoints to achieve good performance. Moreover, based on our complexity analysis in Appendix A.3, the computational cost of this selection algorithm is comparable to training the surrogate model for $E$ epochs. Before starting generation for a specific class $c$, we calculate the average gradient $\bar{\nabla}_{\theta} \ell_c(X_c ; \theta_e^{\mathcal{T}})$ across each of the $K$ retained checkpoints for each class. This process is similar in computational load to training the model for $K$ epochs on the class c of the original dataset and storing $K$ corresponding averaged gradients. All these calculations are performed offline before data generation.
>
> As for the diversity guidance, **the similarity calculation for all generated samples is not computationally demanding** because it is performed in the latent code space of the diffusion model rather than the original image space. **The dimension of the latent code is only 4096**.
>
> More evaluations of the data generation time of our IGD methods can be found in our response A4 to the Reviewer brCT.
>
> ---
> **Q3 (@Weakness3): I would suggest revising the statement that this method is "training-free".**
>
> **A3:** Thank you for highlighting the need for greater clarity regarding this statement. The use of "training-free" was meant to emphasize our proposition that introducing informative guidance is effective for leveraging a well-trained diffusion model in DD tasks. However, we acknowledge that the term "training-free" may be ambiguous. Therefore, we remove the term in line 082 and revise "training-free guidance" in line 137 to "guided-diffusion generation".

---

> ### Author Response · Authors · 2024-11-25
>
> Dear Reviewer hMXA,
>
> Thank you for your positive assessment of our work. We sincerely appreciate your thoughtful comments and constructive feedback on our submission! Below, we briefly summarize our key responses:
>
> - The derivation from stepwise to epochwise in Eq. (4) to Eq. (5) is clarified.
> - Several notational issues in the math part are resolved.
> - The computational and storage costs with a detailed analysis are reported.
> - Ambiguous statements regarding the method being "training-free" are revised.
>
> Could you please review our responses and let us know if you have any further questions? Your feedback is invaluable to us!
>
> Best Regards

---

> > ### Comment · Reviewer_hMXA · 2024-11-26
> > **Thanks**
> >
> > Thanks for the rebuttal and my questions have been largely answered. I maintain my score.

---

> ### Author Response · Authors · 2024-11-26
>
> Dear Reviewer hMXA,
>
> Thank you very much for your positive feedback on our responses! We especially appreciate your attention to the details and the mathematical formulation of our paper. The revisions based on your feedback certainly enhanced the clarity of the submission.
>
> Best regards.

---

### Official Review · Reviewer_brCT · 2024-11-02

**Soundness:** 3
**Presentation:** 3
**Contribution:** 2
**Rating:** 6
**Confidence:** 4

**Summary:**

This paper proposes Influence-Guided Diffusion (IGD) for dataset distillation. IGD solves the problem of poor performance and high resource costs of existing methods at high resolution. IGD proposes a training-free sampling framework which can be used in pretrained diffusion models to generate training-effective data. Extensive experiments show IGD achieves state-of-the-art performance in distilling ImageNet datasets.

**Strengths:**

1. IGD is a training-free framework that can be easily used in any pretrained diffusion models.
2. The target this paper hopes to solve is clear, and the proposed methods solve the problem of data influence and diversity constraint theoretically.
3. The performance improvement of IGD used in DiT and Minimax finetuned DiT is obvious.
4. The ablation study is adequate, including all proposed methods and hyperparameters.

**Weaknesses:**

1. In Table.1, the compared methods are missing like latest method RDED [1] mentioned in section 4.1.
2. Although the proposed method IGD is training-free for diffusion models. It requires training a model to collect the surrogate checkpoints used in Eq. 7. The time consumption should be listed as the paper emphasizes efficiency.
3. The model used in Eq.7 is ConvNet-6. If we change the model like for a bigger one Swin Transformer, will the performance better? Or this model choice is relatively insensitive?
4. Is IGD can be used in other efficient diffusion sampling strategy like DPM [2] solvers?
5. The generation time should be compared between IGD and other methods like current SOTA RDED [1].

Reference:

[1]. Sun P, Shi B, Yu D, et al. On the diversity and realism of distilled dataset: An efficient dataset distillation paradigm[C]//Proceedings of the IEEE/CVF Conference on Computer Vision and Pattern Recognition. 2024: 9390-9399.

[2]. Lu C, Zhou Y, Bao F, et al. Dpm-solver: A fast ode solver for diffusion probabilistic model sampling in around 10 steps[J]. Advances in Neural Information Processing Systems, 2022, 35: 5775-5787.

**Questions:**

Please see weeknesses.

---

> ### Author Response · Authors · 2024-11-20
> **Response to Reviewer brCT (1/2)**
>
> Thank you for your insightful questions and constructive suggestions. Specifically, we followed your suggestion to leverage the DPM solver in our method, which **reduced sample generation time by 50%** with **negligible performance degradation**. We answer you questions as below.
>
> ---
> **Q1 (@Weakness1): The compared methods, such as the latest RDED, are missing in Table 1.**
>
> **A1:** Thank you for pointing this out. We compare our method with RDED, and a state-of-the-art model-inversion-based method CDA [1], on the ImageNette, under two evaluation protocols below:
>
> **Hard-label** evaluation protocol (our default):
> | Test (IPC=50)       | CDA  | RDED | DiT-IGD | Minimax-IGD |
> |:-----------:|:----:|:----:|:-------:|:-----------:|
> | ConvNet-6   | 37.5 | 65.2 |   80.9  |     **82.3**    |
> | ResNetAP-10 | 37.9 | 75.2 |   81.0  |     **82.3**    |
> | ResNet-18   | 38.5 | 75.5 |   81.0  |     **82.0**    |
>
>
> **Soft-label** evaluation protocol (RDED's & CDA's default):
> | Test (IPC=50)       | CDA  | RDED | DiT-IGD | Minimax-IGD |
> |:-----------:|:----:|:----:|:-------:|:-----------:|
> | ConvNet-6   | 78.6 | 81.1 |   **87.8**  |     87.0    |
> | ResNetAP-10 | 80.8 | 83.2 |   86.6  |     **87.0**    |
> | ResNet-18   | 81.8 | 85.0 |   87.6  |     **87.8**    |
>
> The results indicate that **our method achieves superior performance under both evaluation protocols**, despite being primarily designed for the hard-label protocol by default.
>
>
> [1] Yin, Zeyuan, Zhiqiang, Shen. Dataset Distillation via Curriculum Data Synthesis in Large Data Era. Transactions on Machine Learning Research, 2024.
>
> ---
> **Q2(@Weakness3): How robust is the influence guidance calculation across different architectures?**
>
> **A2:** **We have already evaluated the impact of using different-size models as surrogates for influence guidance calculation in Section 4.3**, titled "Cross-Architecture Robustness of Influence Guidance," **and in Table 4**. The experimental results show that the effectiveness of influence guidance is insensitive to the choice of surrogate models.
>
> Your suggestion to use a Swin Transformer, e.g., a non-CNN architecture, as a surrogate is valuable for a more comprehensive robustness evaluation. As a reference, we provid below **the cross-architecture performance using Swin Transformer checkpoints to compute influence guidance** for generating an IPC=50 synthetic dataset over ImageNette:
> | Test Model    | DiT-IGD | Minimax-IGD |
> |---------------|---------|-------------|
> | ConvNet-6     | 78.3    | 79.2        |
> | ResNetAP-10   | 80.6    | 80.8        |
> | ResNet-18     | 79.6    | 80.2        |
>
> From the results above and Table 4, we observe: 1) **the influence guidance effectiveness is generally insensitive to the surrogate model used**; and 2) **using a more complex model as the surrogate tends to slightly underperform compared to simpler models**. Notably, the second observation aligns with findings from gradient-matching-based or training-trajectory-matching-based DD methods which also utilize gradient information. We hypothesize that complex, high-performance models as surrogates might inject less generalizable "short-cut" features into the synthetic data, leading to reduced performance.

---

> ### Author Response · Authors · 2024-11-20
> **Response to Reviewer brCT (2/2)**
>
> **Q3(@Weakness4): Can IGD be used with other efficient diffusion sampling strategies like DPM solvers?**
>
> **A3:** Following your advice, we achieved a **significant 50% reduction in average sampling time** by applying the DPM solver to our IGD method (from 8.2 s to 4.3 s on an RTX 4090).
>
> We used the official implementation of the DPM solver with the default 20 sampling steps. Notably, we observed **negligible performance degradation** or even **slight improvement** with fewer sampling steps. Below, we compare the average performance using DDIM with 50 steps and the DPM solver with 20 steps for distilling ImageNette with IPC=50:
> |      Solver    |   DiT   | DiT-IGD | Minimax | Minimax-IGD |
> |:--------:|:-------:|:-------:|:-------:|:-----------:|
> | DDIM-50  |   75.4  |   80.9  |   77.7  |     82.1    |
> | DPM-20   | 74.1 (&darr;1.3) | **81.9 (&uarr;1.0)** | 76.4 (&darr;1.3) | 80.5 (&darr;1.6) |
>
> We will include and further extend this useful supplementary evaluation in the later revised version of our paper and introduce the corresponding implementation with the DPM solver in our released code.
>
> ---
> **Q4(@Weakness2&5):  The time consumption for training surrogate checkpoints and the generation time of IGD against other methods like RDED should be compared.**
>
> **A4:** Thank you for pointing out the need for greater statement clarity and your constructive suggestions. First, we want to clarify that our method's "training-free" claim means we do not require retraining the diffusion model, as done in Minimax. **Rather than purely focusing on efficiency**, we regard our influence-guided method as **paving a new way for using diffusion for DD tasks by designing effective guidance to improve training efficacy**.
>
> Regarding your suggestion to compare time consumption with SOTA methods like RDED and model-inversion-based methods, all require a well-trained surrogate model for generating synthetic data and predicting soft labels. For example, to distil the ImageNet datasets, these methods need to train a ResNet model on the entire original dataset for over 100 epochs. In contrast, our method only requires training a simpler model, such as ConvNet, for 50 epochs.
>
> For instance generation time, we acknowledge that **RDED is inherently more efficient than diffusion-based methods for dataset distillation**. RDED uses a strategy similar to core-set selection methods to choose informative patches from real images and generate data by stitching them together. This  enables RDED to create a surrogate dataset with IPC=50 for ImageNette/ImageWoof in just a few minutes. By comparison, our method takes approximately 69 minutes with DDIM-50 and 35 minutes with DPM-20, respectively, when generating all classes without using parallel workers.
>
> However, RDED’s reliance on core-patch selection also limits its ability to synthesize new content of distilled data, whereas our guided diffusion method can. This is reflected in our superior performance, especially in settings where only hard labels are used. Therefore, we believe exploring guided diffusion methods remains a promising direction for dataset distillation.

---

> ### Author Response · Authors · 2024-11-25
>
> Dear Reviewer brCT,
>
> We sincerely appreciate the time and effort you have dedicated to reviewing our submission. As the discussion deadline approaches, we would like to provide a summary of our responses and updates:
>
> - Leveraging the DPM solver achieving a **50% reduction in sample generation time** with **negligible performance degradation**.
> - Our method still performs better than RDED in Table 1's comparison.
> - Insensitivity is observed when using the Swin Transformer as a surrogate.
> - Detailed time comparison with RDED are reported.
>
> Would you mind checking our responses and confirming if you have any additional questions? We welcome any further comments and discussions!
>
> Best Regards

---

### Official Review · Reviewer_Xr47 · 2024-11-03

**Soundness:** 3
**Presentation:** 3
**Contribution:** 3
**Rating:** 6
**Confidence:** 4

**Summary:**

The work proposes a guidance scheme for dataset distillation with two main contributions. The first is to do gradient matching between sampled data with the training data, and the second is to add diversity constraints among samples inside a class.The experimental results show clear improvement over other baselines

**Strengths:**

1. The motivation is reasonable
2. The paper is well written
3. The performance is significant

**Weaknesses:**

1. The performance is still far from full dataset.
2. Lack of diversity measurement experiments
3. The design of equation (7) lacks clarification.
4. The application of the work seems to be not flexible. From my understanding, according to one architecture, there will be a need for one-time distillation. Is it possible to have one time distillation and use that distilled datasets to validate across models? I can see Table 4 for the robustness between models, yet the performance is not the same for the used models for guidance. This results in the concern in the application in reality due to computational exhibitions.

**Questions:**

1. The performance is still very far from the original datasets. What is the least IPC to achieve similar performance with full data?
2. Which experiments show an improvement in diversity? The diversity should be measured in terms of FID/Recall values.
3. The equation (7) uses cosine similarity instead of product; is it purely due to experimental results or based on some other hypothesis?
4. How will the work be performed on different tasks apart from classification?

---

> ### Author Response · Authors · 2024-11-20
> **Response to Reviewer Xr47 (1/2)**
>
> Thank you for your positive feedback! Your valuable question and suggestions regarding the evaluation of distribution diversity led new insights on supporting our influence-guided strategy for dataset distillation (DD). Below are our repones to your questions.
>
> ---
> **Q1(@Question1): The performance is still very far from the original datasets. What is the least IPC to achieve similar performance with full data?**
>
> **A1:** As a reference, our Minimax-IGD method achieves **90.8% test accuracy** on a ResNetAP-10 model with IPC=400 (**approximately 31% of the original dataset size**) on the ImageNette dataset under RDED's evaluation protocol. This shows a 3.8% performance gap compared to the test accuracy on the full original dataset (94.6%). Further increasing the IPC only results in marginal improvements.
>
> Empirically, the 3.8% error is largely due to "hard samples" that also result in relatively high test loss for a model trained on the full dataset, indicating these are marginal instances within the authentic data distribution. Due to the inherent objective of denoising diffusion models, synthetic data tends to be sampled from high-probability regions, resulting in poor coverage of these marginal instances. Moreover, as suggested by the definition of averaged gradient $\bar{\nabla}_{\theta} \ell_c(X_c ; \theta_e^{\mathcal{S}})$ (line 180), the influence guidance tends to contribute less to the influence promotion over marginal instances. Addressing this limitation will be a key focus of our future research.
>
> ---
> **Q2(@Question2): Which experiments show an improvement in diversity? The diversity should be measured in terms of FID/Recall values.**
>
> **A2:** In Section 4.5 and **Figure 3**, we provided t-SNE visualizations comparing the distributions of data generated by two baseline methods (DiT and Minimax) and our IGD-based methods with IPC=100. The figure shows that integrating **IGD enhances diversity and alignment** with the original dataset, supported by lower Wasserstein distances to the original dataset.
>
> Our further experiments followed your suggestion reveal that "**focusing solely on diversity or simple alignment with the original dataset is insufficient for optimal effectiveness in DD scenarios**". We compare the **FID scores** and **coverage** of surrogate datasets (IPC=100) generated for ImageWoof by different methods,
> |            Metric           |   DiT   | DiT-IGD | Minimax  | Minimax-IGD | Random  |
> |:---------------------------:|:-------:|:-------:|:--------:|:-----------:|:-------:|
> | **FID**                     |  81.1   |  75.9   |   80.1   |     76.4    | **54.1**|
> | **Coverage (%)**            |  65.4   |  68.1   |   66.5   |     67.2    | **72.3**|
> | **Accuracy (%)**            |  62.3   |  70.6   |   67.4   |   **72.1**  |  63.6   |
>
> Coverage was assessed based on whether each original data point had a nearest neighbor in the surrogate dataset within a given threshold (e.g., 300 in the Inception V3 latent space). For fairness, we excluded data selected by the Random method from the original dataset during coverage calculation.
>
> From the results, although **the randomly selected dataset has the lowest FID and highest coverage, its performance was the worst**. Similarly, while Minimax-IGD has worse FID and coverage than DiT-IGD, it performed better. These findings suggest that our diversity-constraint influence-guided objective is a more effective measure for DD than relying solely on distribution alignment.

---

> ### Author Response · Authors · 2024-11-20
> **Response to Reviewer Xr47 (2/2)**
>
> **Q3(@Question3): Equation (7) uses cosine similarity instead of dot product; is it purely due to experimental results or based on some other hypothesis?**
>
> **A3:** As stated in lines 205-207, we replace the dot product with cosine similarity to primarily stabilize the magnitude of the influence guidance signal. Together with the dynamic scale factor $\rho_t$ defined in Eq. 8, **this replacement allows our method to achieve good performance with the minimal tuning of the influence factor $k$** when using different surrogate models or diffusion samplers (as reported in Table 4 and our response A3@Reviewer-brCT).
>
> Moreover, since Eq. 7 involves checkpoint parameters retained from different training stages, we empirically observed that using the dot product for gradient similarity causes loss from earlier checkpoints to dominate the influence guidance. Cosine similarity alleviates this issue effectively.
>
> ---
> **Q4(@Question4): How will the work be performed on different tasks apart from classification?**
>
> **A4:** As defined in Section 2.1, our work currently focuses on dataset distillation for image classification tasks. Given the limited time during the rebuttal stage, we respectfully provide evaluations centred on our primary focus and comparisons with related work of similar scope. We recognize the potential of this question and plan to explore the applicability of our approach to other tasks in future work.
>
> ---
> **Q5(@Weakness4): Is it possible to have a one-time distillation and use that distilled datasets to validate across models?**
>
> **A5:** Thank you for raising this important question, which relates to a fundamental criterion in DD research. The community often refers to this as **the cross-architecture or unseen-architecture generalization capability of distilled datasets**. This is one of our key motivations for introducing the guided-diffusion paradigm, allowing us to formulate the DD problem as learning a training-effective conditional distribution of the authentic distribution, thereby mitigating the distribution shift issues faced by earlier DD methods (line 108).
>
> As noted in the implementation details (line 317), our method **uses ConvNet-6 as the surrogate model for calculating influence guidance during one-time distillation by default**. The results reported in Tables 1-3 are based on this default setting. These results demonstrate the strong unseen-architecture generalization of our one-shot synthetic dataset across both CNN and Transformer models.

---

> ### Author Response · Authors · 2024-11-25
>
> Dear Reviewer Xr47,
>
> We sincerely appreciate the time and effort you have dedicated to reviewing our submission, as well as your positive feedback on our work! Below is a summary of our responses and updates:
>
> - A discussion regarding the least IPC needed for the full dataset performance is provided.
> - Experiments measuring diversity with FID and distribution coverage have been included.
> - The rationale behind using cosine similarity in Equation (7) is further clarified.
> - The feasibility of using a one-time distilled dataset across unseen models is elucidated.
>
> We would greatly appreciate it if you could check our responses and let us know if there are any additional questions. Your feedback is invaluable to us!
>
> Best Regards

---

> > ### Comment · Reviewer_Xr47 · 2024-11-28
> > **I confirm to read rebuttals**
> >
> > Dear authors,
> >
> > I confirm to read your rebuttals carefully. My concern is still in its generalizability. I can acknowledge the importance of the research direction. Yet, there are still concerns
> >
> > I concern that the datasets are generated to fit to only one dataset and one task. If so, the application of this work is very limited. Whenever a new task comes, the data distillation needs to be done one which is actually time-consuming even more than the training time of the task with full data. The classification does not take much time to train, but for more expensive task like generative task, not so sure if the distilled dataset can work. This will limit the application of the work.
> >
> > Even the case that it is specifically designed for one task, it still can not achieve the SOTA without the constraint on the number of instances.

---

> > > ### Author Response · Authors · 2024-11-28
> > >
> > > Dear Reviewer Xr47,
> > >
> > > Thank you for your thoughtful review! We appreciate your insightful questions, which have inspired us to reflect on future directions in dataset distillation.
> > >
> > > To the best of our knowledge, **all current DD methods require generating a new distilled dataset for each unseen task-specific dataset**. However, your question raises an important point: similar to transfer learning for downstream tasks, is it possible to leverage knowledge from previously distilled datasets to improve distillation efficiency or performance on new tasks? We plan to explore this idea in future work.
> > >
> > > We also acknowledge that achieving scalability for complex generative tasks is challenging but valuable. Unlike classification, which benefits from clear decision boundaries, improving the data efficiency of generative models remains an open problem. A recent work [1] has proposed a coreset selection method based on distribution alignment, which shows moderate performance on smaller datasets. However, the scalability of such methods for more complex tasks remains an open challenge.
> > >
> > > We share your concern about distilled datasets not matching full dataset performance, which is a common issue in methods including ours and state-of-the-art baslines we compared, e.g., RDED and SRe$^2$L based on model-inversed information. While these methods can achieve remarkable performance at lower IPCs, a gap remains due to limitations in diversity and marginal case coverage. As we noted earlier, addressing this gap will be a focus of our future work.
> > >
> > > Thank you once again for your insightful feedback, which has greatly contributed to the direction of our ongoing and future work. If you have any further thoughts or concerns regarding our work or the future of this field, we would be grateful for your continued guidance.
> > >
> > > Best Regards.
> > >
> > > [1] Yize Li, Yihua Zhang, Sijia Liu, Xue Lin. Pruning then Reweighting: Towards Data-Efficient Training of Diffusion Models. CoRR abs/2409.19128 (2024)

---

### Official Review · Reviewer_rhZM · 2024-11-04

**Soundness:** 3
**Presentation:** 3
**Contribution:** 3
**Rating:** 6
**Confidence:** 4

**Summary:**

This paper addresses the challenges of dataset distillation, which aims to create compact yet effective datasets for training larger original datasets. Existing methods often face limitations when dealing with large, high-resolution datasets due to high resource costs and suboptimal performance, largely due to sample-wise optimizations in the pixel space. To overcome these challenges, the authors propose framing dataset distillation as a controlled diffusion generation task, leveraging the capabilities of diffusion generative models to learn target dataset distributions and generate high-quality data tailored for training.

The authors introduce the Influence-Guided Diffusion (IGD) sampling framework, which generates training-effective data without retraining the diffusion models. This is achieved by establishing a connection between the goal of dataset distillation and the trajectory influence function, using this function as an indicator to guide the diffusion process toward promoting data influence and enhancing diversity. The proposed IGD method is shown to significantly improve the training performance of distilled datasets and achieves state-of-the-art results in distilling ImageNet datasets. Notably, the method reaches an impressive performance of 60.3% on ImageNet-1K with IPC (Images Per Class) set to 50.

**Strengths:**

1. The paper is easy to read and understand.
2. IGD appears to be superior to existing diffusion model-based approaches.

**Weaknesses:**

1. In the introduction, the authors introduce the concept of Influence-Guided without clearly explaining what "Influence" entails or why it is used for guidance. The motivation is not well established. While Figure 1 effectively shows performance, adding an additional subfigure to illustrate the motivation or highlight differences from previous methods might be more valuable.

2. The primary contribution of the authors is the proposal of a train-free diffusion framework for Dataset Distillation. While train-free approaches are common in the AIGC field, how does the proposed method differ from existing ones?

3. The experiments only report results on ImageNet, without including results on classic datasets such as CIFAR-10 and CIFAR-100.

**Questions:**

As shown in Table 5, the main contributions of the authors include the proposed influence guidance and deviation guidance. What is the relationship between these contributions and the "train-free" concept? Notably, even when these components are excluded from Equation 9, good results are still achieved.

---

> ### Author Response · Authors · 2024-11-20
> **Response to Reviewer rhZM (1/1)**
>
> Thank you for your instructive feedback and valuable suggestions. Below are our responses to your questions.
>
> ---
> **Q1(@Weakness1): The introduction lacks a clear explanation of what "influence" entails or why it is used for guidance. Adding a figure to illustrate the motivation or differences from previous methods could be helpful.**
>
> **A1:** **We have uploaded a revised version of the paper incorporating your constructive feedback**. In lines 59-80 of the Introduction, we added **a brief explanation of the influence function** and reorganized the text to highlight its role in addressing the inherent challenges posed by the abstract nature of our training-effective condition for diffusion generation. Additionally, in the newly added Figure 5, **we provide an intuitive illustration of the IGD sampling framework**, contrasting it with the vanilla diffusion sampling method.
>
> We respectfully look forward to your feedback on the updated content.
>
> ---
> **Q2(@Weakness2): How does the proposed training-free diffusion framework for dataset distillation differ from existing training-free approaches in AIGC?**
>
> **A2:** This is a key question related to our motivation for proposing the influence-guided paradigm for diffusion generation. In common diffusion-based AIGC applications, users primarily control data generation through text prompts (line 134). While this enables some degree of content specification, **systematically defining a diverse set of high-quality prompts** to effectively guide diffusion models in generating both diverse and training-effective data for dataset distillation (DD) **remains an abstract challenge without a structured optimization methodology**.
>
> To address these challenges, we identify the influence function as an effective metric that measures the compatibility of generated data with generalized training-effective conditions. Building on this insight, **our proposed IGD method is the first to systematically optimize data influence to generate high-quality surrogate datasets for DD tasks**.
>
> ---
> **Q3(@Weakness3): The experiments only report results on ImageNet, without including results on classic datasets such as CIFAR-10 and CIFAR-100.**
>
> **A3:** We provide a comparison of our DiT-IGD method with other state-of-the-art DD methods designed for distilling high-resolution, large-scale datasets, including RDED [1] and SRe$^2$L [2], on CIFAR-10 as a reference.
> | IPC | SRe$^2$L  |  RDED   | DiT-IGD |
> |:---:|:------:|:-------:|:-------:|
> |  10 |  29.3  | **37.1** |  35.8   |
> |  50 |  45.0  |  62.1   | **63.5** |
>
> Our method achieves comparable performance with RDED on CIFAR-10. However, for the primary focus of large-scale DD tasks, our method attains significantly better performance, as shown in Tables 2-3 and our response A1@Reviewer-brCT.
>
> [1] Peng Sun, Bei Shi, Daiwei Yu, Tao Lin. On the Diversity and Realism of Distilled Dataset: An Efficient Dataset Distillation Paradigm. CVPR 2024
>
> [2] Zeyuan Yin, Eric P. Xing, Zhiqiang Shen. Squeeze, Recover and Relabel: Dataset Condensation at ImageNet Scale From A New Perspective. NeurIPS 2023
>
> ---
> **Q4(@Question1): What is the relationship between the two proposed guidance and the "train-free" concept? Notably, even when these components are excluded from Equation 9, good results are still achieved.**
>
> **A4:** In this work, we frame the DD problem as learning a training-effective conditional distribution of the authentic distribution, thereby addressing the distribution shift issues encountered by earlier DD methods (line 108). As mentioned in **A3**, addressing this problem requires tackling the abstract nature of a generalized training-effective condition. To systematically optimize data influence and generate high-quality surrogate datasets for DD tasks, we introduce two effective guidance to steer the diffusion sampling process that provides influence promotion and diversity enhancement.
>
> We respectfully disagree with the statement that "good results are still achieved even when these guidance components are excluded." Our comparative results in Tables 1 and 2 over ImageNet datasets show **significant improvements over the two baselines**, DiT and Minimax, when integrating our influence guidance and deviation guidance, **as also acknowledged by reviewers brCT, hMXA, and Qttw**. For instance, IGD enhances the average performance of DiT by 6.6% and provides a 5.1% boost for Minimax on ImageWoof when IPC ≥ 50. Furthermore, our ablation study in Table 5 shows the contributions of the two guidance components.

---

> > ### Comment · Reviewer_rhZM · 2024-11-22
> > **Further Discussion**
> >
> > I appreciate the authors' response. While some of my comments were addressed, others, in my view, require further discussion.
> >
> > 1. Regarding Q3, the authors provided results only on the CIFAR-10 dataset. The results indicate that with IPC=10, the performance of DATM is 66.8, whereas IGD achieves only 35.8. This demonstrates a significant performance gap, raising concerns about the efficiency of IGD.
> >
> > 2. My concern persists regarding why the method is considered **train-free** and how it aligns with the target domain. Additionally, the relationship between performance improvement and being train-free remains unclear. I believe the authors may not have fully understood my main concern. While I acknowledge that ablation experiments in every paper aim to demonstrate the validity of the approach in terms of performance, this is not the aspect I am focusing on.
> >
> > Further clarification on these points would be appreciated.

---

> > > ### Author Response · Authors · 2024-11-25
> > >
> > > Dear Reviewer rhZM,
> > >
> > > Thank you for your continued engagement and valuable feedback on our submission! Below is a summary of our responses to your further questions:
> > >
> > > - Our method achieves comparable lossless performance to DATM on CIFAR-10.
> > > - Further clarifications and revisions regarding the "training-free" term is provided.
> > >
> > > We would greatly appreciate it if you could review our additional responses and let us know if you have further questions or concerns. We look forward to your further feedback!
> > >
> > > Best Regards

---

> > > > ### Comment · Reviewer_rhZM · 2024-11-25
> > > > **Last question**
> > > >
> > > > Most of my questions have been addressed by the authors. However, one remaining point raises concerns about the development of the DD field. Specifically, there appear to be two dominant categories of methods: (1) DATM, which demonstrates strong performance on small datasets (commonly IPC = 10, 50), and (2) high-resolution-oriented methods, which have gained traction more recently. While some of these high-resolution methods essentially combine a single high-resolution image with multiple lower-resolution images, these two approaches seem fundamentally contradictory.
> > > >
> > > > This leads to my main concern: how do methods like DATM perform on high-definition (HD) datasets? Conversely, methods tailored for high-resolution data often struggle with lower-resolution datasets, and most of these approaches lack reported results on small-scale datasets. Is there a unified method capable of achieving strong performance across both low-resolution and high-resolution datasets?
> > > >
> > > > I would encourage the authors to construct a more comprehensive benchmark that incorporates both low-resolution and high-resolution datasets, as this would significantly enhance the generalizability and robustness of the evaluation. If such a benchmark is developed, I will be inclined to increase my overall score.

---

> ### Author Response · Authors · 2024-11-23
> **Further discussion with Reviewer rhZM**
>
> Thank you for your reply! We believe your further questions are related to **Q3 (@Weakness3)** and **Q4 (@Question1)** from our previous response. Below, we provide additional discussion on these two questions.
>
> ---
> **Further discussion on Q3(@Weakness3):**
>
> We would like to respectfully recall that the significant progress of early DD frameworks in distilling small datasets like CIFARs was acknowledged (lines 40-44). However, the limitations identified in lines 45-50 (e.g, cost increase with data dimension or high-frequency features) **restrict their applicability to more practical, high-resolution, large-scale datasets** (e.g., 224×224 ImageNets), which is the focus of our research (line 301). These observations motivated us to leverage diffusion models to generate **near-real but training-effective** synthetic data.
>
> Thank you for mentioning DATM, a representative training trajectory matching DD method that achieves lossless performance on CIFARs. While DATM significantly outperforms our method or RDED (which aims to generate near-real distilled data) at IPC=10 for CIFAR-10, the test performance of 66.8% still has a significant gap compared to using the full dataset (e.g., 84.8%). More importantly, the core contribution of DATM is to emphasize the use of late trajectory matching to **generate hard samples with fewer distilled features (closer to real images)** for **lossless distillation** when IPC is high.
>
> Due to limited time during the rebuttal phase, we continue to use CIFAR-10 as a reference here. The following results demonstrate that **our method achieves approximately lossless performance with a 20% compression ratio (IPC=1000), comparable to DATM**:
>
> | Test   | ConvNet | ResNet-18 | VGG-11  |
> |:-------:|:-------:|:---------:|:-------:|
> | DATM    |   85.5  |  87.2   |  84.6   |
> | DiT-IGD |   84.6  |  85.8   |   84.0    |
>
> Implementation details will be released later in our official code.
>
>
> ---
> **Further discussion on Q4 (@Question1):**
>
> We apologize for not fully understanding your concern earlier. The term "training-free" was meant to emphasize the practicality of introducing informative guidance with a well-trained diffusion model in DD tasks, rather than enhancing distribution alignment through retraining, as done in Minimax.
>
> Inspired by the remarkable ability of denoising diffusion models to learn a parameterized distribution that approximates the authentic distribution (line 144), a natural extension is to define **a conditional distribution aligned with specific task requirements**. A line of work known as "training-free guided-diffusion generation" based on energy-based models (EBM) (line 138) has verified the effectiveness of this strategy for guiding diffusion models to meet requirements that are hard to be described by text (e.g., segmentation-guided generation). This motivated us to adopt guided-diffusion frameworks to steer the diffusion model in generating data under training-effective conditions. Furthermore, the improvement achieved by our influence-guided generation over the Minimax method also verifies that **our method can effectively complement DD frameworks like Minimax which require retraining the diffusion model**.
>
> We acknowledge that the term "training-free" may have caused ambiguity. To prevent any misunderstanding, we have removed the term in line 082 and revised "training-free guidance" in line 137 to "guided-diffusion generation" in the updated version of the paper.
>
> We hope this response addresses your concern and further clarifies our motivation. If any questions remain, we would appreciate any specific suggestions or further questions you may have to help improve our clarity.

---

> ### Author Response · Authors · 2024-11-25
>
> Dear Reviewer rhZM,
>
> Thank you very much for the effort and continued attention you have given to our submission!
>
> In Table 1, we have presented **two representative DD methods**, e.g., Distribution Matching (DM) and IDC, **primarily designed for distilling small datasets but can also distil larger datasets**, such as 10-class ImageNet subsets, with acceptable computational resources. However, both methods demonstrated poor performance. Moreover, given that these methods require tens of hours to distil even 10-class subsets of ImageNet, we found it impractical and unnecessary to include them for the distillation of ImageNet-1K in Table 2.
>
> Additionally, while trajectory matching-based methods like DATM can achieve superior performance on small datasets such as CIFAR, **their computational and time costs are prohibitive for distilling 224×224 ImageNet datasets**. These costs primarily stem from:
>
> - Constructing an expert trajectory pool requires training dozens of surrogate models on the full dataset, which is extremely time-consuming for large datasets.
> - The trajectory-matching loss used to optimize distilled data involves unrolling the computation graph of multiple gradient descent steps, which is highly memory-intensive (≥ 100 GB) and time-consuming.
>
> Due to these challenges, we considered employing these methods for distilling high-resolution datasets to be intractable and impractical. Given the limited time remaining in the discussion phase (less than 48 hours), we respectfully emphasize **the difficulty of supplementing such a benchmark at this stage**, as well as **its divergence from the focus of our submission**.
>
> If our responses have addressed most of your concerns regarding our submission, we would respectfully request your kind consideration of increasing the rating. We truly appreciate your positive feedback on our submission!
>
> Best regards.

---

> ### Author Response · Authors · 2024-11-28
> **Supplement Benchmark Testing High-Resolution-Oriented Methods on Small Datasets**
>
> Dear Reviewer rhZM,
>
> As highlighted in our previous responses, **adapting most low-resolution-oriented DD methods to distil high-resolution datasets presents scalability issues**. We found it intractable to report their performance on ImageNet datasets evaluated in our submission. For reference, we have provided benchmarks for other state-of-the-art baselines (including two low-resolution-oriented methods DM and IDC) in Tables 1 and 2, which have been largely well-received by the other reviewers.
>
> We agree with your valuable suggestion to **test high-resolution-oriented methods on small datasets** to better assess the generalizability of the methods. As per your recommendation, we have established a comprehensive benchmark comparing the performance of ConvNet using state-of-the-art high-resolution-oriented methods, including SRe$^2$L [1] and RDED, on CIFAR-10 and CIFAR-100 at various IPCs.
>
> CIFAR-10:
> | Method  | DM   | DATM  | SRe$^2$L | RDED  | DiT-IGD |
> |---------|------|-------|-------|-------|---------|
> | IPC 50 (1%)| 63.1 | 76.1  | 43.2  | 68.4  | 66.8    |
> | IPC 500 (10%)| 74.3 | 83.5  | 55.3  | 78.1  | 82.6    |
> | IPC 1000 (20%)| 79.2 | 85.5  | 57.1  | 79.8  | 84.6    |
>
> CIFAR-100:
> | Method   | DM   | DATM  | SRe$^2$L | RDED  | DiT-IGD |
> |----------|------|-------|-------|-------|---------|
> | IPC 10 (2%)| 29.7 | 47.2  | 24.5  | 46.4  | 45.8    |
> | IPC 50 (10%)| 43.6 | 55.0  | 45.2  | 51.5  | 53.9    |
> | IPC 100 (20%)| 47.1 | 57.5  | 46.6  | 52.6  | 55.9    |
>
>
> The results show that our method outperforms SRe$^2$L and RDED in most scenarios. Additionally, it achieves nearly lossless performance similar to DATM (e.g., at a 20% compression ratio). These findings, combined with our method's outstanding performance on ImageNet datasets, suggest that **our approach is a unified solution that performs well across low-resolution and high-resolution datasets**.
>
> Thank you for your constructive suggestion! We have included these evaluations in Appendix A5 of the current revised paper. We would appreciate your feedback on the benchmark evaluation we have provided.
>
> Best Regards.
>
> [1] Zeyuan Yin, Eric P. Xing, Zhiqiang Shen. Squeeze, Recover and Relabel: Dataset Condensation at ImageNet Scale From A New Perspective. NeurIPS 2023

---

> ### Author Response · Authors · 2024-12-01
>
> Dear Reviewer rhZM,
>
> Thank you very much for your proactive attitude and continued engagement with our submission! We deeply appreciate your constructive suggestions, which have significantly helped improve our work.
>
> In response to your previous comment titled "Last Question," we have added **a benchmark evaluating our method and other high-resolution-oriented methods on smaller datasets (CIFAR-10 & CIFAR-100)**. We have **included this valuable benchmark in Appendix A5** of the revised paper.  Together with our method's strong performance on ImageNet datasets, this suggests that our approach provides a unified solution effective across both low-resolution and high-resolution datasets.
>
> With the discussion period extended, we look forward to any further valuable feedback you may have on the additional evaluation. Any other suggestions or thoughts on future directions for this field from you are also invaluable to us!
>
> Best Regards.

---

> > ### Comment · Reviewer_rhZM · 2024-12-02
> > **Good benckmark**
> >
> > Thanks for the authors' response and I have raised my score.

---

> ### Author Response · Authors · 2024-12-02
>
> Dear Reviewer rhZM,
>
> We are truly grateful for your kind recognition of our submission and the additional content! Your constructive feedback and suggestions have significantly improved our work!
>
> We deeply appreciate your thorough review and proactive attitude to enhancing the paper, both of which are invaluable to the research community.
>
> Wishing you all the best!
>
> Best Regards.

---

### Author Response · Authors · 2024-11-28
**Summary of Major Revisions in the Updated Submission based on Reviewer Feedback**

We deeply appreciate the reviewers for their thoughtful and constructive comments, which have greatly contributed to improving our work. We have thoroughly revised the paper to address the reviewers' concerns, as outlined in our responses to their questions. Below is a summary of the major revisions in the current submission.

**Summary of Major Revisions:**
1. We added **an intuitive illustration**, contrasting it with the vanilla diffusion sampling method in **Figure 5**. (A1@Reviewer rhZM)
2. In lines 59-80, we included a brief explanation of the trajectory influence function and reorganized the text to emphasize its role in achieving training-effectiveness in diffusion generation. (A1@Reviewer rhZM)
3. We included **additional comparisons on CIFAR-10 and CIFAR-100 in Appendix A5** to demonstrate the versatility of our method in distilling smaller datasets. (Last Question@Reviewer rhZM)
4. We added **comparisons with recent diffusion-based approaches in Appendix A6**, showing the continued superior performance of our method. (A3@Reviewer Qttw)
5. We evaluated our methods **using the DPM solver for diffusion generation in Appendix A7**, demonstrating a 50% reduction in sample generation time with negligible performance degradation. (A3@Reviewer brCT)
6. We expanded **the analysis of distribution diversity and coverage in Appendix A8 using FID and coverage metrics**. (A2@Reviewer Xr47)
7. We addressed several notational issues and removed the term "training-free" in line 082 as well as revised "training-free guidance" in line 137 to "guided-diffusion generation" to avoid ambiguity. (A1&A3@Reviewer hMXA)

We sincerely appreciate the reviewers for their insightful feedback, which has greatly contributed to refining our work. The revisions, including additional analyses, new baselines, and expanded experiments, have significantly improved the clarity and comprehensiveness of the paper. We welcome any further discussions or feedback and are happy to provide additional clarifications or materials as needed.

---

### Meta-Review · Area_Chair_JvuM · 2024-12-18

**Metareview:**

The paper introduce a new method called Influence-Guided Diffusion for dataset distillation. IGD leverages a training-free sampling framework with pretrained diffusion models to generate training-effective data, incorporating gradient matching and diversity constraints for improved performance. The method demonstrates state-of-the-art results on ImageNet distillation.

The authors have made substantial revisions to enhance the clarity and depth of the paper. Key updates include the addition of an intuitive illustration comparing their method to the standard diffusion sampling approach, as well as a reorganization of the text to highlight the role of the trajectory influence function in improving training effectiveness. Additional experimental results on CIFAR-10 and CIFAR-100 were incorporated to demonstrate the method’s versatility, alongside comparisons with recent diffusion-based approaches. These revisions enhance the comprehensiveness and clarity of the paper.

**Additional Comments On Reviewer Discussion:**

Reviewers generally were satisfied with the authors' response. The reviewer highlighted challenges like scalability, task-specific dataset generation, and performance gaps, which could be common issues in dataset distillation, not unique to this paper. Despite these inherent challenges, the paper makes valuable advancements and have meaningful contributions to the field.

---

### Decision · Program_Chairs · 2025-01-22

Accept (Poster)